# Voint Cloud: Multi-View Point Cloud Representation for 3D Understanding

**Abdullah Hamdi**             **Silvio Giancola**             **Bernard Ghanem**

**King Abdullah University of Science and Technology (KAUST), Thuwal, Saudi Arabia**
{abdullah.hamdi, silvio.giancola, bernard.ghanem}@kaust.edu.sa

## Abstract

Multi-view projection methods have demonstrated promising performance on 3D understanding tasks like 3D classification and segmentation. However, it remains unclear how to combine such multi-view methods with the widely available 3D point clouds. Previous methods use unlearned heuristics to combine features at the point level. To this end, we introduce the concept of the multi-view point cloud (Voint cloud), representing each 3D point as a set of features extracted from several view-points. This novel 3D Voint cloud representation combines the compactness of 3D point cloud representation with the natural view-awareness of multi-view representation. Naturally, we can equip this new representation with convolutional and pooling operations. We deploy a Voint neural network (VointNet) to learn representations in the Voint space. Our novel representation achieves state-of-the-art performance on 3D classification, shape retrieval, and robust 3D part segmentation on standard benchmarks ( ScanObjectNN, ShapeNet Core55, and ShapeNet Parts).[1]

## 1 Introduction

A fundamental question in 3D computer vision and computer graphics is how to represent 3D data (Mescheder et al., 2019; Qi et al., 2017a; Maturana & Scherer, 2015). This question becomes particularly vital given how the success of deep learning in 2D computer vision has pushed for the wide adoption of deep learning in 3D vision and graphics. In fact, deep networks already achieve impressive results in 3D classification (Hamdi et al., 2021), 3D segmentation (Hu et al., 2021), 3D detection (Liu et al., 2021a), 3D reconstruction (Mescheder et al., 2019), and novel view synthesis (Mildenhall et al., 2020). 3D computer vision networks either rely on *direct* 3D representations, *indirect* 2D projection on images, or a mixture of both. *Direct* approaches operate on 3D data commonly represented with point clouds (Qi et al., 2017a), meshes (Feng et al., 2019), or voxels (Choy et al., 2019). In contrast, *indirect* approaches commonly render multiple 2D views of objects or scenes (Su et al., 2015), and process each image with a traditional 2D image-based architecture. The human visual system is closer to such a multi-view *indirect* approach for 3D understanding, as it receives streams of rendered images rather than explicit 3D data.

Tackling 3D vision tasks with *indirect* approaches has three main advantages: **(i)** mature and transferable 2D computer vision models (CNNs, Transformers, *etc.* ), **(ii)** large and diverse labeled image datasets for pre-training (*e.g.* ImageNet (Russakovsky et al., 2014)), and **(iii)** the multi-view images give context-rich features based on the viewing angle, which are different from the geometric 3D neighborhood features. Multi-view approaches achieve impressive performance in 3D shape classification and segmentation (Wei et al., 2020; Hamdi et al., 2021; Dai & Nießner, 2018). However, the challenge with the multi-view representation (especially for dense predictions) lies in properly aggregating the per-view features with 3D point clouds. The appropriate aggregation is necessary to obtain representative 3D point

---

[1]The code is available at https://github.com/ajhamdi/vointcloud

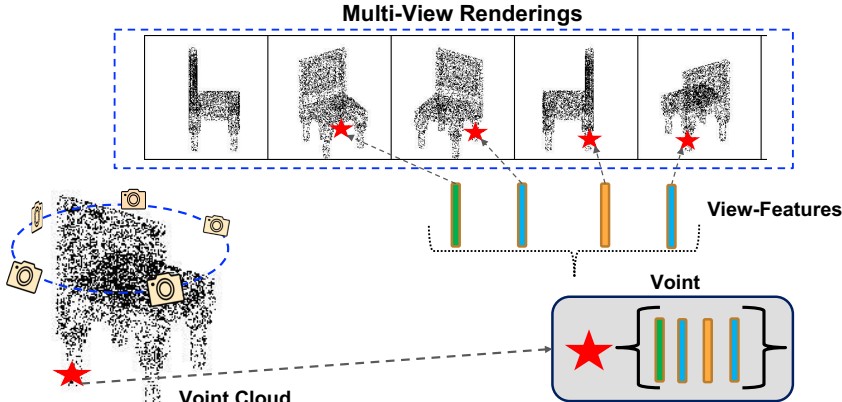

Figure 1: **3D Voint Clouds.** We propose the multi-view point cloud (Voint cloud), a novel 3D representation that is compact and naturally descriptive of view projections of a 3D point cloud. Each point in the 3D cloud is tagged with a Voint, which accumulates view-features for that point. Note that not all 3D points are visible from all views. The set of Voints constructs a Voint cloud.

clouds with a single feature per point suitable for typical point cloud processing pipelines. Previous multi-view works rely on heuristics (*e.g.* average or label mode pooling) after mapping pixels to points (Kundu et al., 2020; Wang et al., 2019a), or multi-view fusion with voxels (Dai & Nießner, 2018). Such setups might not be optimal for a few reasons. **(i)** Such heuristics may aggregate information of misleading projections that are obtained from arbitrary view-points. For example, looking at an object from the bottom and processing that view independently can carry wrong information about the object's content when combined with other views. **(ii)** The views lack geometric 3D information.

To this end, we propose a new hybrid 3D data structure that inherits the merits of point clouds (*i.e.* compactness, flexibility, and 3D descriptiveness) and leverages the benefits of rich perceptual features of multi-view projections. We call this new representation multi-view point cloud (or *Voint cloud*) and illustrate it in Figure 1. A Voint cloud is a set of Voints, where each Voint is a set of view-dependent features (view-features) that correspond to the same point in the 3D point cloud. The cardinality of these view-features may differ from one Voint to another. In Table 1, we compare some of the widely used 3D representations and our Voint cloud representation. Voint clouds inherit the characteristics of the parent explicit 3D point clouds, which facilitates learning Voint representations for a variety of vision applications (*e.g.* point cloud classification and segmentation). To deploy deep learning on the new Voint space, we define basic operations on Voints, such as pooling and convolution. Based on these operations, we define a practical way of building Voint neural networks that we dub *VointNet*. VointNet takes a Voint cloud and outputs point cloud features for 3D point cloud processing. We show how learning this Voint cloud representation leads to strong performance and gained robustness for the tasks of 3D classification, 3D object retrieval, and 3D part segmentation on standard benchmarks like ScanObjectNN (Uy et al., 2019), and ShapeNet (Chang et al., 2015).

**Contributions: (i)** We propose a novel multi-view 3D point cloud representation (denoted as *Voint cloud*), which represents each point (namely a *Voint*) as a set of features from different view-points. **(ii)** We define pooling and convolutional operations at the Voint level to construct a Voint Neural Network (VointNet) capable of learning to aggregate information from multiple views in the Voint space. **(iii)** Our VointNet reaches state-of-the-artperformance on several 3D understanding tasks, including 3D shape classification, retrieval, and robust part segmentation. Further, VointNet achieves robustness improvement to occlusion and rotation.

| 3D Representation | Explicitness | View-Based | Main Use | 3D Expressiveness |
|---|---|:---:|:---:|:---:|
| Point Clouds | Explicit | ✗ | 3D Understanding | Medium |
| Multi-View Projections | Implicit | ✓ | 3D Understanding | Low |
| Voxels | Explicit | ✗ | 3D Understanding | Medium |
| Mesh | Explicit | ✗ | 3D Modeling | High |
| NeRFs | Implicit | ✓ | Novel View Synthesis | Medium |
| **Voint Clouds (ours)** | Explicit | ✓ | 3D Understanding | Medium |

Table 1: **Comparison of Different 3D Representations**. We compare some of the widely used 3D representations to our proposed Voint cloud. Note that our Voint cloud shares the view-dependency of NeRFs (Mildenhall et al., 2020) while inheriting the merits of 3D point clouds.

## 2 Related Work

**Learning on 3D Point Clouds.** 3D point clouds are widely used for 3D representation in computer vision due to their compactness, flexibility, and because they can be obtained naturally from sensors like LiDAR and RGBD cameras. PointNet (Qi et al., 2017a) paved the way as the first deep learning algorithm to operate directly on 3D point clouds. It computes point features independently and aggregates them using an order-invariant function like max-pooling. Subsequent works focused on finding neighborhoods of points to define point convolutional operations (Qi et al., 2017b; Wang et al., 2019c; Li et al., 2018; Han et al., 2019). Several recent works combine point cloud representations with other 3D modalities like voxels (Liu et al., 2019b; You et al., 2018) or multi-view images (Jaritz et al., 2019). We propose a novel Voint cloud representation for 3D shapes and investigates novel architectures that aggregate view-dependent features at the 3D point level.

**Multi-View Applications.** The idea of using 2D images to understand the 3D world was initially proposed in 1994 by Bradski *et. al.* (Bradski & Grossberg, 1994). This intuitive multi-view approach was combined with deep learning for 3D understanding in MVCNN (Su et al., 2015). A line of works continued developing multi-view approaches for classification and retrieval by improving the aggregation of the view-features from each image view (Kanezaki et al., 2018; Esteves et al., 2019; Cohen & Welling, 2016; Wei et al., 2020; Hamdi et al., 2021). In this work, we fuse the concept of multi-view into the 3D structure itself, such that every 3D point would have an independent set of view-features according to the view-points available in the setup. Our Voints are aligned with the sampled 3D point cloud, offering a compact representation that allows for efficient computation and memory usage while maintaining the view-dependent component that facilitates view-based learning for vision.

**Hybrid Multi-View with 3D Data.** On the task of 3D semantic segmentation, a smaller number of works tried to follow the multi-view approach (Dai & Nießner, 2018; Kundu et al., 2020; Wang et al., 2019a; Kalogerakis et al., 2017; Jaritz et al., 2019; Liu et al., 2021b; Lyu et al., 2020). A problem arises when combining view features to represent local points/voxels while preserving local geometric features. These methods tend to average the view-features (Kundu et al., 2020; Kalogerakis et al., 2017), propagate the labels only (Wang et al., 2019a), learn from reconstructed points in the neighborhood (Jaritz et al., 2019), order points on a single grid (Lyu et al., 2020), or combine the multi-view features with 3D voxel features (Dai & Nießner, 2018; Hou et al., 2019). To this end, our proposed VointNet operates on the Voint cloud space while preserving the compactness and 3D descriptiveness of the original point cloud. VointNet leverages the power of multi-view features with learned aggregation on the view-features applied to each point *independently*.

## 3 Methodology

The primary assumption in our work is that surface 3D points are spherical functions, *i.e.* their representations depend on the viewing angles observing them. This condition contrasts with most 3D point cloud processing pipelines that assume a view-independent representation of 3D point clouds. The full pipeline is illustrated in Figure 2.

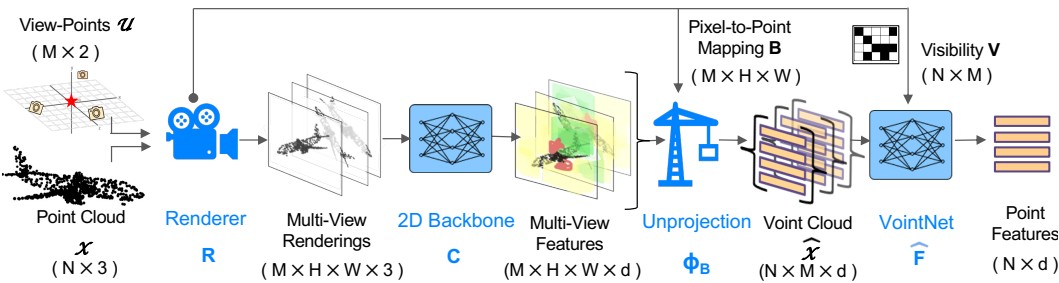

Figure 2: **Learning from Voint Clouds.** To construct a 3D Voint cloud $\widehat{\mathcal{X}}$, a renderer $\mathbf{R}$ renders the point cloud $\mathcal{X}$ from view-points $\mathcal{U}$ and image features are extracted from the generated images via a 2D backbone $\mathbf{C}$. The image features are then unprojected to the Voint cloud by $\Phi_{\mathbf{B}}$ and passed to VointNet $\widehat{\mathbf{F}}$. To learn both $\mathbf{C}$ and $\widehat{\mathbf{F}}$, a 3D loss on the output points is used with an optional auxiliary 2D loss on $\mathbf{C}$.

## 3.1 3D Voint Cloud

**From Point Clouds to Voint Clouds.** A 3D point cloud is a compact 3D representation composed of sampled points on the surface of a 3D object or a scene and can be obtained by different sensors like LiDAR (Chen et al., 2017) or as a result of reconstruction (Okutomi & Kanade, 1993). Formally, we define the coordinate function for the surface $g_{\mathrm{s}}(\mathbf{x}) : \mathbb{R}^3 \to \mathbb{R}$ as the Sign Distance Function (SDF) in the continuous Euclidean space (Park et al., 2019; Mescheder et al., 2019). The 3D iso-surface is then defined as the set of all points $\mathbf{x}$ that satisfy the condition $g_{\mathrm{s}}(\mathbf{x}) = 0$. We define a surface 3D point cloud $\mathcal{X} \in \mathbb{R}^{N \times 3}$ as a set of $N$ 3D points, where each point $\mathbf{x}_i \in \mathbb{R}^3$ is represented by its 3D coordinates $(x_i, y_i, z_i)$ and satisfies the iso-surface condition as follows: $\mathcal{X} = \left\{ \mathbf{x}_i \in \mathbb{R}^3 \ \mid \ g_{\mathrm{s}}(\mathbf{x}_i) = 0 \right\}_{i=1}^{N}$. In this work, we aim to fuse the view-dependency to 3D point. Inspired by NeRFs (Mildenhall et al., 2020), we assume that surface points also depend on the view direction from which they are being observed. Specifically, there exists a continuous implicit spherical function $\mathbf{g}(\mathbf{x}, \mathbf{u}) : \mathbb{R}^5 \to \mathbb{R}^d$ that defines the features of each point $\mathbf{x}$ depending on the view-point direction $\mathbf{u}$. Given a set of $M$ view-point directions $\mathcal{U} \in \mathbb{R}^{M \times 2}$, a Voint $\widehat{\mathbf{x}} \in \mathbb{R}^{M \times d}$ is a set of $M$ view-dependent features of size $d$ for the sphere centered at point $\mathbf{x}$ as follows.

$$\widehat{\mathbf{x}}_i = \left\{ \mathbf{g}\left(\mathbf{x}_i, \mathbf{u}_j\right) \in \mathbb{R}^d \ \mid \ \mathbf{x}_i \in \mathcal{X} \right\}_{j=1}^{M} \tag{1}$$

The Voint cloud $\widehat{\mathcal{X}} \in \mathbb{R}^{N \times M \times d} = \{\widehat{\mathbf{x}}_i\}_{i=1}^{N}$ is the set of all $N$ Voints $\widehat{\mathbf{x}}_i$ corresponding to the parent point cloud $\mathcal{X}$. Note that we typically do not have access to the underlying implicit function $\mathbf{g}$ and we approximate it with the following three steps.

**1- Multi-View Projection.** As mentioned earlier, a Voint combines multiple view-features of the same 3D point. These view-features come from a multi-view projection of the points by a point cloud renderer $\mathbf{R} : \mathbb{R}^{N \times 3} \to \mathbb{R}^{M \times H \times W \times 3}$ that renders the point cloud $\mathcal{X}$ from multiple view-points $\mathcal{U}$ into $M$ images of size $H \times W \times 3$. In addition to projecting the point cloud into the image space, $\mathbf{R}$ defines the index mapping $\mathbf{B} \in \{0, .., N\}^{M \times H \times W}$ between each pixel to the N points and background it renders. Also, $\mathbf{R}$ outputs the visibility binary matrix $\mathbf{V} \in \{0, 1\}^{N \times M}$ for each point from each view. Since not all points appear in all the views due to pixel discretization, the visibility score $\mathbf{V}_{i,j}$ defines if the Voint $\widehat{\mathbf{x}}_i$ is visible in the view $\mathbf{u}_j$. The matrix $\mathbf{B}$ is crucial for unprojection, while $\mathbf{V}$ is needed for defining meaningful operations on Voints.

**2- Multi-View Feature Extraction.** The rendered images are processed by a function $\mathbf{C} : \mathbb{R}^{M \times H \times W \times 3} \to \mathbb{R}^{M \times H \times W \times d}$ that extracts image features, as shown in Figure 2. If $\mathbf{C}$ is the identity function, all the view-features would typically the RGB value of the corresponding point. However, the $\mathbf{C}$ function can be a 2D network dedicated to the downstream task and can extract useful global and local features about each view.

**3- Multi-View Unprojection.** We propose a module $\Phi_{\mathbf{B}} : \mathbb{R}^{M \times H \times W \times d} \to \mathbb{R}^{N \times M \times d}$ that unprojects the 2D features from each pixel to be 3D view-features at the corresponding voint. Using the mapping $\mathbf{B}$ created by the renderer, $\Phi_{\mathbf{B}}$ forms the Voint cloud features $\widehat{\mathcal{X}}$.

To summarize, the output Voint cloud is described by Eq (1), where $\mathbf{g}\left(\mathbf{x}_i, \mathbf{u}_j\right) = \Phi_{\mathbf{B}}\big(\mathbf{C}\left(\mathbf{R}\left(\mathcal{X}, \mathbf{u}_j\right)\right)\big)_i$ and the features are only defined for a view $j$ of Voint $\widehat{\mathbf{x}}_i$ if $\mathbf{V}_{i,j} = 1$.

## 3.2 Operations on 3D Voint Clouds

We show in the **Appendix** that a functional form of max-pooled individual view-features of a set of angles can approximate any function in the spherical coordinates. We provide a theorem that extends PointNet's theorem of point cloud functional composition (Qi et al., 2017a) and its Universal Approximation to spherical functions underlying Voints. Next, we define a set of operations on Voints as building blocks for Voint neural networks (VointNet).

**VointMax.** We define VointMax as max-pooling on the visible view-features along the views dimension of the voint $\widehat{\mathbf{x}}$. For all $i \in 1, 2, ..., N$ and $j \in 1, 2, ..., M$,

$$\text{VointMax}(\widehat{\mathbf{x}}_i) = \max_j \ \widehat{\mathbf{x}}_{i,j}, \quad \text{s.t.} \ \ \mathbf{V}_{i,j} = 1 \tag{2}$$

**VointConv.** We define the convolution operation $h_{\mathrm{V}} : \ \mathbb{R}^{N \times M \times d} \to \mathbb{R}^{N \times M \times d'}$ as any learnable function that operates on the Voint space with shared weights on all the Voints and has the view-features input size $d$ and outputs view-features of size $d'$ and consists of $l_V$ layers. A simple example of this VointConv operation is the shared MLP applied *only* on the visible view-features. We provide further details for such operations in Section 4.2, which result in different non-exhaustive variants of VointNet.

## 3.3 Learning on 3D Voint Clouds

**VointNet.** The goal of the VointNet model is to obtain multi-view point cloud features that can be subsequently used by any point cloud processing pipeline. The VointNet module $\widehat{\mathbf{F}} : \mathbb{R}^{N \times M \times d} \to \mathbb{R}^{N \times d}$ is defined as follows.

$$\widehat{\mathbf{F}}(\widehat{\mathcal{X}}) = h_{\mathrm{P}}\left(\text{VointMax}\left(h_{\mathrm{V}}(\widehat{\mathcal{X}})\right)\right), \tag{3}$$

where $h_{\mathrm{P}}$ is any point convolutional operation (*e.g.* shared MLP or EdgeConv). VointNet $\widehat{\mathbf{F}}$ transforms the individual view-features using the learned VointConv $h_{\mathrm{V}}$ before VointMax is applied on the view-features to obtain point features.

**VointNet Pipeline for 3D Point Cloud Processing.** The full pipeline is described in Figure 2. The loss for this pipeline can be described as follows:

$$\underset{\boldsymbol{\theta}_{\mathbf{C}}, \boldsymbol{\theta}_{\widehat{\mathbf{F}}}}{\arg \min} \ \sum_i^N L \left(\widehat{\mathbf{F}}\big(\Phi_{\mathbf{B}}\big(\mathbf{C}\left(\mathbf{R}\left(\mathcal{X}, \mathcal{U}\right)\right)\big)\big)_i \ , \ \mathbf{y}_i\right), \tag{4}$$

where $L$ is a Cross-Entropy (CE) loss defined on all the training points $\mathcal{X}$, and $\{y_i\}_{i=1}^N$ defines the labels of these points. The other components $(\mathbf{R}, \Phi_{\mathbf{B}}, \mathcal{U}, \mathbf{C})$ are all defined before. The weights to be jointly learned are those of the 2D backbone $(\boldsymbol{\theta}_{\mathbf{C}})$ and those of the VointNet $(\boldsymbol{\theta}_{\widehat{\mathbf{F}}})$ using the same 3D loss. An auxiliary 2D loss on $\boldsymbol{\theta}_{\mathbf{C}}$ can be optionally added for supervision at the image level. For classification, the entire object can be treated as a single Voint, and the global features of each view would be the view-features of that Voint. We analyze different setups in detail in Section 6.

## 4 Experiments

### 4.1 Experimental Setup

**Datasets.** We benchmark VointNet on the challenging and realistic ScanObjectNN dataset for 3D point cloud classification (Uy et al., 2019). The dataset has three variants, includes background and occlusion, and has 15 categories and 2,902 point clouds. For the shape retrieval task, we benchmark on ShapeNet Core55 as a subset of ShapeNet (Chang et al., 2015). The dataset consists of 51,162 3D mesh objects labeled with 55 object classes. We follow the MVTN's setup (Hamdi et al., 2021) in sampling 5,000 points from each mesh object to obtain point cloud. On the other hand, for the task of shape part segmentation,

| Method | Data Type | Classification Overall Accuracy | | |
| --- | --- | --- | --- | --- |
| | | OBJ_BG | OBJ_ONLY | Hardest |
| PointNet (Qi et al., 2017a) | Points | 73.3 | 79.2 | 68.0 |
| SpiderCNN (Xu et al., 2018) | Points | 77.1 | 79.5 | 73.7 |
| PointNet ++ (Qi et al., 2017b) | Points | 82.3 | 84.3 | 77.9 |
| PointCNN (Li et al., 2018) | Points | 86.1 | 85.5 | 78.5 |
| DGCNN (Wang et al., 2019c) | Points | 82.8 | 86.2 | 78.1 |
| SimpleView (Goyal et al., 2021) | M-View | - | - | 79.5 |
| BGA-DGCNN (Uy et al., 2019) | Points | - | - | 79.7 |
| BGA-PN++ (Uy et al., 2019) | Points | - | - | 80.2 |
| MVTN (Hamdi et al., 2021) | M-View | 92.6 | 92.3 | 82.8 |
| VointNet (ours) | Voints | **93.7** | **94.0** | **85.4** |

Table 2: **3D Point Cloud Classification on ScanObjectNN**. We report the accuracy of VointNet in 3D point cloud classification on three different variants of ScanObjectNN (Uy et al., 2019). **Bold** denotes the best result in its setup. Note that the *Hardest* variant includes rotated and translated objects, which highlights the benefits of Voints on challenging scenarios.

we test on ShapeNet Parts (Yi et al., 2016), a subset of ShapeNet (Chang et al., 2015) that consists of 16,872 point cloud objects from 16 categories and 50 parts. For occlusion robustness, we follow MVTN (Hamdi et al., 2021) and test on ModelNet40 (Wu et al., 2015), which is composed of 40 classes and 12,311 3D objects.

**Metrics.** For 3D point cloud classification, we report the overall accuracy, while shape retrieval is evaluated using mean Average Precision (mAP) over test queries (Hamdi et al., 2021). 3D semantic segmentation is evaluated using mean Intersection over Union (mIoU) on points. For part segmentation, we report Instance-averaged mIoU (Ins. mIoU).

**Baselines.** We include PointNet (Qi et al., 2017a), PointNet++ (Qi et al., 2017b), DGCNN (Wang et al., 2019c), as baselines that use point clouds. We also compare against multi-view classification approaches like MVCNN (Su et al., 2015), SimpleView (Goyal et al., 2021), and MVTN (Hamdi et al., 2021) as baselines for classification and retrieval and adopt some of the multi-view segmentation baselines (*e.g.* Label Fusion (Wang et al., 2019a) and Mean Fusion (Kundu et al., 2020)) for part segmentation.

## 4.2 VOINTNET VARIANTS

VointNet in Eq (3) relies on the VointConv operation $h_V$ as the basic building block. Here, we briefly describe three examples of $h_V$ operations VointNet uses.

**Shared Multi-Layer Perceptron (MLP).** It is the most basic VointConv formulation. For a layer $l$, the features of Voint $i$ at view $j$ are updated to layer $l+1$ as: $\mathbf{h}_{i,j}^{l+1} = \rho \left( \mathbf{h}_{i,j}^{l} \mathcal{W}_\rho \right)$, where $\rho$ is the shared MLP with weights $\mathcal{W}_\rho$ followed by normalization and a nonlinear function (*e.g.* ReLU). This operation is applied on all Voints independently and only involves the visible views-features for each Voint. This formulation extends the shared MLP formulation for PointNet (Qi et al., 2017a) to work on Voints' view-features.

**Graph Convolution (GCN).** We define a fully connected graph *for each Voint* by creating a virtual center node connected to all the view-features to aggregate their information (similar to "cls" token in ViT (Dosovitskiy et al., 2021)). Then, the graph convolution can be defined as the shared MLP (as described above) but on the edge features between all view features, followed by a max pool on the graph neighbors. An additional shared MLP is used before the final output.

**Graph Attention (GAT).** A graph attention operation can be defined just like the GCN operation above but with learned attention weights on the graph neighbor's features before averaging them. A shared MLP computes these weights.

| Results | MVCNN (Su et al., 2015) | RotNet (Kanezaki et al., 2018) | ViewGCN (Wei et al., 2020) | MVTN (Hamdi et al., 2021) | VointNet (ours) |
|---|---|---|---|---|---|
| ShapeNet Retr. mAP | 73.5 | 77.2 | 78.4 | 82.9 | **83.3** |

Table 3: **3D Shape Retrieval**. We report 3D shape retrieval mAP on ShapeNet Core55 (Chang et al., 2015; Sfikas et al., 2017).VointNet achieves state-of-the-art results on this benchmark.

| Method | Data Type | Part Segmentation (Unrotated) | (Rotated) |
|---|---|---|---|
| PointNet (Qi et al., 2017a) | Points | 80.1 | 36.6 ±0.2 |
| DGCNN (Wang et al., 2019c) | Points | 80.1 | 37.1 ±0.2 |
| CurveNet (Xiang et al., 2021) | Points | **84.9** | 32.3 ±0.0 |
| Label Fuse (Wang et al., 2019a) | M-View | 80.0 | 61.4 ±0.2 |
| Mean Fuse (Kundu et al., 2020) | M-View | 77.5 | 62.0 ±0.2 |
| VointNet (ours) | Voints | 81.2 | **62.4** ±0.2 |

Table 4: **Robust 3D Part Segmentation on ShapeNet Parts**. We compare the Inst. mIoU of VointNet against other methods in 3D segmentation on ShapeNet Parts (Yi et al., 2016). At test time, we randomly rotate the objects and report the results over ten runs. Note how VointNet's performance largely exceeds the point baselines in the realistic rotated scenarios, while exceeding multi-view baselines on the unrotated benchmark. All the results are reproduced in our setup.

### 4.3 IMPLEMENTATION DETAILS

**Rendering and Unprojection.** We choose the differentiable point cloud renderer **R** from Pytorch3D (Ravi et al., 2020) in our pipeline for its speed and compatibility with Pytorch libraries (Paszke et al., 2017). We render point clouds on multi-view images with size $224 \times 224 \times 3$. We color the points by their normals' values or keep them white if the normals are not available. Following a similar procedure to (Wei et al., 2020; Hamdi et al., 2021), the view-points setup is randomized during training (using $M = 8$ views) and fixed to spherical views in testing (using $M = 12$ views).

**Architectures.** For the 2D backbone **C**, we use ViT-B (Dosovitskiy et al., 2021) (with pretrained weights from TIMM library (Wightman, 2019)) for classification and DeepLabV3 (Chen et al., 2018) for segmentation. We use the 3D CE loss on the 3D point cloud output and the 2D CE loss when the loss is defined on the pixels. The feature dimension of the VointNet architectures is $d = 64$, and the depth is $l_V = 4$ layers in $h_V$. The main results are based on VointNet (MLP), unless otherwise specified as in Section 6, where we study in details the effect of VointConv $h_V$ and **C**.

**Training Setup.** We train our pipeline in two stages, where we start by training the 2D backbone on the 2D projected labels of the points, then train the entire pipeline end-to-end while focusing the training on the VointNet part. We use the AdamW optimizer (Loshchilov & Hutter, 2017) with an initial learning rate of 0.0005 and a step learning rate schedule of 33.3% every 12 epochs for 40 epochs. The pipeline is trained with one NVIDIA Tesla V100 GPU. We do not use any data augmentation. More details about the training setup (loss and rendering), VointNet, and the 2D backbone architectures can be found in the **Appendix** .

## 5 RESULTS

The main test results of our Voint formulations are summarized in Tables 2,3, 4, and 5. We achieve state-of-the-artperformance in the task of 3D classification, retrieval, and robust 3D part segmentation. More importantly, under the realistic rotated setups of ScanObjectNN and ShapeNet Parts, we improve over 7.2 % Acc. and 25% mIoU respectively compared to point baselines Qi et al. (2017a); Wang et al. (2019c). Following common practice Hamdi et al. (2021), we report the best results out of four runs in benchmark tables, but detailed results are provided in the **Appendix** .

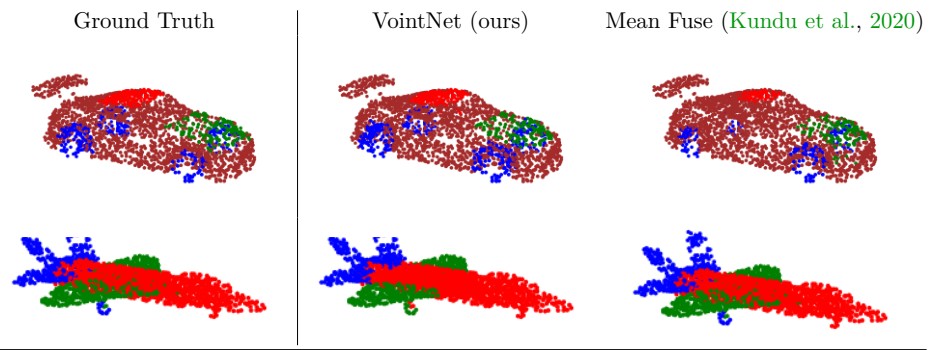

Figure 3: **Qualitative Comparison for Part Segmentation.** We compare our VointNet 3D segmentation predictions to Mean Fuse (Kundu et al., 2020) that is using the same trained 2D backbone. Note how VointNet distinguishes detailed parts (*e.g.* the car window frame).

## 5.1 3D SHAPE CLASSIFICATION

Table 2 reports the classification accuracy on the 3D point cloud classification task on ScanObjectNN Uy et al. (2019). It benchmarks VointNet against other recent and strong baselines Hamdi et al. (2021); Goyal et al. (2021); Hamdi et al. (2021). VointNet demonstrates state-of-the-artresults on all the variants, including the challenging Hardest (PB_T50_RS) variant that includes challenging scenarios of rotated and translated objects. The increase in performance (+2.6%) is significant on this variant, which highlights the benefits of Voints on challenging scenarios, with further affirming results in Section 5.4. We follow exactly the same procedure as in MVTN Hamdi et al. (2021).

## 5.2 3D SHAPE RETRIEVAL

Table 3 benchmarks the 3D shape retrieval mAP on ShapeNet Core55 Chang et al. (2015). VointNet achieves state-of-the-artperformance on ShapeNet Core55. Baseline results are reported from Hamdi et al. (2021).

## 5.3 ROBUST 3D PART SEGMENTATION

Table 4 reports the Instance-averaged segmentation mIoU of VointNet compared with other methods on ShapeNet Parts Yi et al. (2016). Two variants of the benchmark are reported : unrotated normalized setup, and the rotated realistic setup. For the rotated setup, we follow the previous 3D literature Liu et al. (2019a); Hamdi et al. (2021; 2020) by testing the robustness of trained models by perturbing the shapes in ShapeNet Parts with random rotations at test time (ten runs) and report the averages in Table 4. Note VointNet's improvement over Mean Fuse Kundu et al. (2020) and Label Fuse Wang et al. (2019a) on unrotated setup despite that both baselines use the same trained 2D backbone as VointNet. Also, for rotated setups, point methods don't work as well. All the results in Table 4 are reproduced by our code in the same setup (see the code attached in supplementary material). Figure 3 shows qualitative 3D segmentation results for VointNet and Mean Fuse Kundu et al. (2020) as compared to the ground truth.

## 5.4 OCCLUSION ROBUSTNESS

One of the aspects of the robustness of 3D classification models that have been recently studied is their robustness to occlusion, as detailed in MVTN Hamdi et al. (2021). These simulated occlusions are introduced at test time, and the average test accuracy is reported on each cropping ratio. We benchmark our VointNet against recent baselines in Table 5. PointNet Qi et al. (2017a) and DGCNN Wang et al. (2019c) are used as point-based baselines, and MVTN Hamdi et al. (2021) as a multi-view baseline.

| Method | Data Type | Occlusion Ratio | | | | |
|---|---|---|---|---|---|---|
| | | 0 | 0.1 | 0.2 | 0.3 | 0.5 |
| PointNet (Qi et al., 2017a) | Points | 89.1 | 88.2 | 86.1 | 81.6 | 53.5 |
| DGCNN (Wang et al., 2019c) | Points | 92.1 | 77.1 | 74.5 | 71.2 | 30.1 |
| PCT (Guo et al., 2021) | Points | 93.3 | **92.6** | 91.1 | 88.2 | 61.9 |
| MVTN (Hamdi et al., 2021) | M-View | **93.8** | 90.3 | 89.9 | 88.3 | **67.1** |
| VointNet (ours) | Voints | 92.8 | 91.6 | **91.2** | **89.1** | 66.1 |

Table 5: **Occlusion Robustness for 3D Classification.** We report the test accuracy on ModelNet40 (Wu et al., 2015) for different occlusion ratios of the data to measure occlusion robustness of different 3D methods.

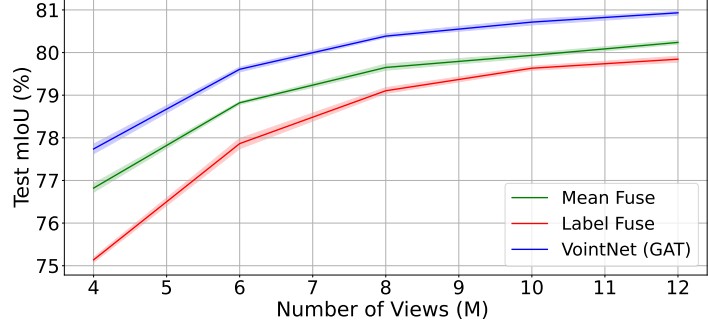

Figure 4: **Effect of the Number of Views.** We plot Ins. mIoU of 3D segmentation *vs.* the number of views (*M*) used in inference on ShapeNet Parts. Note VointNet's consistent improvement over Mean Fuse (Kundu et al., 2020) and Label Fuse (Wang et al., 2019a). Both baselines use the same trained 2D backbone as VointNet and are tested on the same unrotated setup.

## 6 Analysis and Insights

**Number of Views.** We study the effect of the number of views *M* on the performance of 3D part segmentation using multiple views. We compare Mean Fuse (Kundu et al., 2020) and Label Fuse (Wang et al., 2019a) to our VointNet when all of them have the same trained 2D backbone. The views are randomly picked, and the experiments are repeated four times. Ins. mIoU with confidence intervals are shown in Figure 4. We observe a consistent improvement with VointNet over the other two baselines across different numbers of views.

| 2D Backbone | | VointConv | | | Results |
|---|---|---|---|---|---|
| FCN | DeepLabV3 | MLP | GCN | GAT | Inst. mIoU |
| ✓ | - | ✓ | - | - | 78.8 ± 0.2 |
| ✓ | - | - | ✓ | - | 77.6 ± 0.2 |
| ✓ | - | - | - | ✓ | 77.1 ± 0.2 |
| - | ✓ | ✓ | - | - | 80.6 ± 0.1 |
| - | ✓ | - | ✓ | - | 77.2 ± 0.4 |
| - | ✓ | - | - | ✓ | 80.4 ± 0.2 |

Table 6: **Ablation Study for 3D Segmentation**. We ablate different components of VointNet (2D backbone and VointConv choice) and report Ins. mIoU performance on ShapeNet Parts.

**Choice of Backbones.** We ablate the choice of the 2D backbone and the VointConv operation used in VointNet and report the segmentation Ins. mIoU results in Table 6. Note how the 2D backbone greatly affects performance, while the VointConv operation type does not. This ablation highlights the importance of the 2D backbone in VointNet pipeline and motivates the use of the simplest variant of VointNet (MLP). We provide a detailed study of more factors as well as compute and memory costs in the **Appendix** .

## 7  Limitations and Acknowledgments

One aspect limiting the performance of Voints is how well-trained the 2D backbone is for the downstream 3D task. In most cases, the 2D backbone must be pretrained with enough data to learn meaningful information for VointNet. Another aspect that limits the capability of the Voint cloud is how to properly select the view-points for segmentation. Addressing these limitations is an important direction for future work. Also, extending Voint learning on more 3D tasks like 3D scene segmentation and 3D object detection is left for future work.

**Acknowledgments.** This work was supported by the King Abdullah University of Science and Technology (KAUST) Office of Sponsored Research through the Visual Computing Center (VCC) funding and the SDAIA-KAUST Center of Excellence in Data Science and Artificial Intelligence (SDAIA-KAUST AI)

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

# Appendix

## A  Detailed Formulations

### A.1  Toy Example

In the toy 2D example in Figure 5, the center point (represented by a circular function $g$) is viewed from various view-points $u_j$ that are agnostic to the underlying function itself. In many applications, it is desired to have a single feature representing each point in the point cloud. When the projected values of $g$ from these $u_j$ view-points are aggregated together (*e.g.* by max/mean pool) to get a constant representation of that point, the underlying properties of $g$ are lost. We build our Voint representation to keep the structure of $g$ intact by taking the full set $\{(u_j, g(u_j))\}_{j=1}^5$ in learning the aggregations.

### A.2  Functional Form of VointNet

We can look at a simplified setup to decide on the functional form of the deep neural network that operates in the Voint space. In this simplified setup, we consider a 2D example (instead of 3D Voints) and assume that a circular function describes a point at the center. The center point will assume its value according to the angle $u$. The following Theorem 1 proves that for any continuous set function $f$ that operates on any set of $M$ angles $\{u_1, ..., u_M\}$, there exists an equivalent composite function consisting of transformed max-pooled individual view-features. This composition is the functional form we describe later for Voint neural networks

**Theorem 1** *Suppose $f : \mathcal{S} \to \mathbb{R}$ is a continuous set function operating on an angles set $\mathcal{S} = \{u \mid u \in [0, 2\pi]\}$. The continuity of $f$ is based on the Hausdorff distance $d_H$ between two sets of angles, where $d_H(\mathcal{S}, \mathcal{S}') = \max_{u_i' \in \mathcal{S}'} \min_{u_i \in \mathcal{S}} d_A(u_i, u_i')$, and $d_A$ is the smallest positive angle between two angles $d_A(u, u') = \min(|u - u'|, 2\pi - |u - u'|)$. Then, for every $\epsilon > 0$, and $\mathcal{U} = \{u_1, ..., u_M\} \subset \mathcal{S}$, there exists a continuous function $\mathbf{h}$ and a symmetric function $g(u_1, ..., u_M) = \gamma \circ \mathrm{MAX}$, such that:*

$$\left| f(\mathcal{U}) - \gamma\Big( \mathrm{MAX}\big(\mathbf{h}(u_1), \ldots, \mathbf{h}(u_M)\big)\Big) \right| < \epsilon, \tag{5}$$

*where $\gamma$ is a continuous function, and $\mathrm{MAX}$ is an element-wise vector max operator.*

*Proof.* By the continuity of $f$, we take $\delta_\epsilon$ so that $|f(\mathcal{U}) - f(\mathcal{U}')| < \epsilon$ for any $\mathcal{U}, \mathcal{U}' \subset \mathcal{S}$ if $d_H(\mathcal{U}, \mathcal{U}') < \delta_\epsilon$. Define $K = [2\pi/\delta_\epsilon]$, which split $[0, 2\pi]$ into $K$ intervals evenly and define an auxiliary function that maps an angle to the beginning of the interval it lies in:

$$\sigma(u) = \frac{\lfloor Ku \rfloor}{K}$$

Let $\tilde{\mathcal{U}} = \sigma(u) : u \in \mathcal{U}$, then

$$|f(\mathcal{U}) - f(\tilde{\mathcal{U}})| < \epsilon \tag{6}$$

Let $h_k(u) = e^{-d\left(u, \left[\frac{k-1}{K}, \frac{k}{K}\right]\right)}$ be a soft indicator function where $d\left(u, \left[\frac{k-1}{K}, \frac{k}{K}\right]\right) = \min\left(d_A\left(u, \frac{k-1}{K}\right), d_A\left(u, \frac{k}{K}\right)\right)$ is the distance between angle $u$ to interval $\left[\frac{k-1}{K}, \frac{k}{K}\right]$. Let $\mathbf{h}(u) = [h_1(u); \ldots; h_K(u)]$, then $\mathbf{h} : \mathbb{R} \to \mathbb{R}^K$

Let $q_j(u_1, \ldots, u_M) = \max\{h_j(u_1), \ldots, h_j(u_M)\}$, indicating the occupancy of the $j$-th interval by angles in $\mathcal{U}$. Let $\mathbf{q} = [q_1; \ldots; q_K]$, then $\mathbf{q} : [0, 2\pi]^M \to \{0, 1\}^K$ is a symmetric function, indicating the occupancy of each interval by angles in $\mathcal{U}$.

Define $\zeta : \{0, 1\}^K \to \mathcal{S}$ as $\zeta(\mathbf{q}) = \left\{\frac{k-1}{K} : q_k \geq 1\right\}$ which maps the occupancy vector to a set which contains the left end of each angle interval. It is straightforward to show:

$$\zeta(\mathbf{q}(\mathcal{U})) \equiv \tilde{\mathcal{U}} \tag{7}$$

Let $\gamma : \mathbb{R}^K \to \mathbb{R}$ be a continuous function such that $\gamma(\mathbf{q}) = f(\zeta(\mathbf{q}))$ for $\mathbf{q} \in \{0, 1\}^K$. Then from Eq (6) and Eq (7),

$$\begin{aligned} &|\gamma(\mathbf{q}(\mathcal{U})) - f(\mathcal{U})| \\ &= |f(\zeta(\mathbf{q}(\mathcal{U}))) - f(\mathcal{U})| < \epsilon \end{aligned} \tag{8}$$

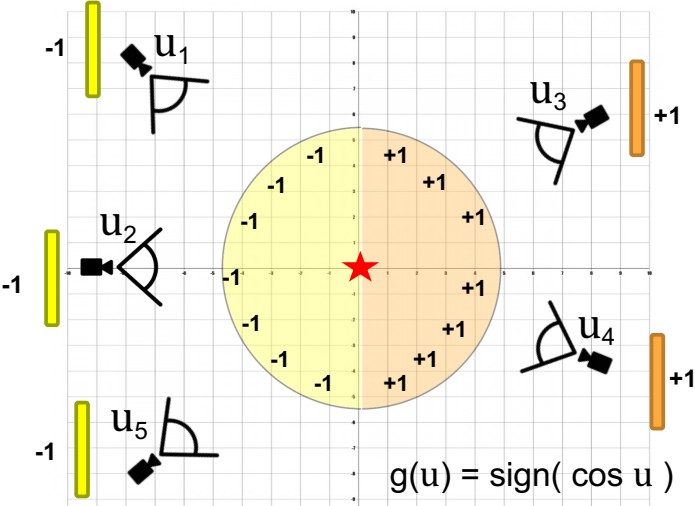

Figure 5: **A Toy 2D Example of Voints.** Voints assume view-dependency for every 3D point. Here, we look at a single 2D point at the center with a circular function $g(u) = \text{sign}(\cos u)$ from five arbitrary view-points $\{u_j\}_{j=1}^5$. Trying to reduce $g$ to a single value based on $u_j$ projections undermines the underlying structure of $g$. We take the full set $\{(u_j, g(u_j))\}_{j=1}^5$ as a representation of $g$ and learn a set function $f$ on these view-features for a more informative manner of representation aggregation.

Note that $\gamma(\mathbf{q}(\mathcal{U}))$ can be rewritten as follows:

$$\begin{aligned} \gamma(\mathbf{q}(\mathcal{U})) &= \gamma(\mathbf{q}(u_1, \ldots, u_M)) \\ &= \gamma(\text{MAX}(\mathbf{h}(u_1), \ldots, \mathbf{h}(u_M))) \\ &= (\gamma \circ \text{MAX})(\mathbf{h}(u_1), \ldots, \mathbf{h}(u_M)) \end{aligned} \tag{9}$$

Since $\gamma \circ \text{MAX}$ is a symmetric function and from Eq (8) and Eq (9), we reach to the main result in Eq (5). This concludes the proof. $\qquad\square$

### A.3 3D VOINT CLOUD

**Plenoptic and Spherical Coordinate Functions.** The Plenoptic function was first introduced by McMillan and Bishop (McMillan & Bishop, 1995) in 1995 as a general function that describes the visible world. The Plenoptic function $P$ is a continuous spherical function that describes the visibility at any Euclidean 3D point in space $(V_x, V_y, V_x)$ when looking into any direction $(\theta, \phi)$ across wavelength $\lambda$ at time $t$. It is defined as $p = P(\theta, \phi, \lambda, V_x, V_y, V_x, t)$. Such a remarkable and compact formulation covers all the images observed as just samples of the function $P$. For fixed time and wavelength, the reduced Plenoptic function $P$ becomes $p = P(\theta, \phi, V_x, V_y, V_x,)$ which can describe any field in 3D space. This shortened formulation is what Neural Radiance Fields (NeRFs) (Mildenhall et al., 2020; Pumarola et al., 2021; Martin-Brualla et al., 2021) try to learn with MLPs to describe the radiance and RGB values in the continuous Euclidean space with a dependency on the view direction $(\theta, \phi)$. In the same spirit of the Plenoptic function and NeRFs, the Voint cloud representation relies on the viewing angles $(\theta, \phi)$ to define the view-features. The problem with the plenoptic functions $P$, and subsequently NeRFs, is that they are very high dimensional, and any attempt to densely represent the scene with discrete and fixed data will cause memory and compute issues (Yu et al., 2021; Pumarola et al., 2021). Unlike NERFs (Mildenhall et al., 2020) that define dense 3D volumes, we focus only on the surface of the 3D shapes with our Voint clouds representation. Our Voints are in the order of the sampled point cloud, offering a compact representation that allows for efficient computation and memory while maintaining the view-dependent component that facilitates view-based learning.

**From Point Clouds to Voint Clouds.** Implicit representation of 3D surfaces typically aims to learn an implicit function $g_s(\mathbf{x}) : \mathbb{R}^3 \to \mathbb{R}$ that define the Sign Distance Function

(SDF) or the occupancy in the continuous Euclidean space ([Park et al., 2019](); [Mescheder et al., 2019]()). The 3D iso-surface is then defined as the set of all points $\mathbf{x}$ that satisfy the condition $g_{\mathrm{s}}(\mathbf{x}) = 0$ (assuming $g_{\mathrm{s}}(\mathbf{x})$ as SDF hereafter). We define a surface 3D point cloud $\mathcal{X} \in \mathbb{R}^{N \times 3}$, as a set of $N$ 3D points, where each point $\mathbf{x}_i \in \mathbb{R}^3$ is represented by its 3D coordinates $(x_i, y_i, z_i)$ and satisfy the iso-surface condition as follows.

$$\mathcal{X} = \left\{ \mathbf{x}_i \in \mathbb{R}^3 | g_{\mathrm{s}}(\mathbf{x}_i) = 0 \right\}_{i=1}^{N} \tag{10}$$

Here, we assume that surface points also depend on the view direction from which they are being observed. Specifically, there exists a continuous implicit spherical function $\mathbf{g}(\mathbf{x}, \mathbf{u})$ : $\mathbb{R}^5 \to \mathbb{R}^d$ that defines the features at each point $\mathbf{x}$ depending on the view direction $\mathbf{u}$. Given a set of $M$ view-point directions $\mathcal{U} \in \mathbb{R}^{M \times 2}$, a Voint $\widehat{\mathbf{x}} \in \mathbb{R}^{M \times d}$ is a set of $M$ view-dependent features of size $d$ for the sphere centered at point $\mathbf{x}$. The Voint cloud $\widehat{\mathcal{X}} \in \mathbb{R}^{N \times M \times d}$ is the set of all $N$ Voints $\widehat{\mathbf{x}}$.

$$\begin{aligned} \widehat{\mathbf{x}}_i &= \left\{ \mathbf{g}\left(\mathbf{x}_i, \mathbf{u}_j\right) \in \mathbb{R}^d \ \mid \ \mathbf{x}_i \in \mathcal{X} \right\}_{j=1}^{M} \\ \widehat{\mathcal{X}} &= \left\{ \widehat{\mathbf{x}}_i \in \mathbb{R}^{M \times d} \right\}_{i=1}^{N} \end{aligned} \tag{11}$$

Note that we typically do not have access to the underlying implicit function $\mathbf{g}$ and we approximate it by 2D projection, feature extraction, and then un-projection as we show next.

**1- Multi-View Projection.** As mentioned earlier, a Voint combines multiple view-features of the same 3D point. These view-features come from a multi-view projection of the points by a point cloud renderer $\mathbf{R} : \mathbb{R}^{N \times 3} \to \mathbb{R}^{M \times H \times W \times 3}$ that renders the point cloud $\mathcal{X}$ from multiple view-points $\mathcal{U}$ into $M$ images of size $H \times W \times 3$. In addition to projecting the point cloud into the image space, $\mathbf{R}$ defines the mapping $\mathbf{B} \in \{0, .., N\}^{M \times H \times W}$ between each pixel to the N points and background it renders. Also, $\mathbf{R}$ outputs the visibility binary matrix $\mathbf{V} \in \{0, 1\}^{N \times M}$ for each point from each view. Since not all points appear in all the views due to pixel discretization, the visibility score $\mathbf{V}_{i,j}$ defines if the Voint $\widehat{\mathbf{x}}_i$ is visible in the view $\mathbf{u}_j$. The matrix $\mathbf{B}$ is crucial for unprojection, while $\mathbf{V}$ is needed for defining meaningful operations on Voints.

**2- Multi-View Feature Extraction.** The rendered images are processed by a function $\mathbf{C} : \mathbb{R}^{M \times H \times W \times 3} \to \mathbb{R}^{M \times H \times W \times d}$ that extracts image features. If $\mathbf{C}$ is the identity function, all the view-features would be identical for each Voint (typically the RGB value of the corresponding point). However, the $\mathbf{C}$ function can be a 2D network dedicated to the downstream task and can extract useful global and local features about each view.

**3- Multi-View Unprojection.** We propose a module $\Phi_{\mathbf{B}} : \mathbb{R}^{M \times H \times W \times d} \to \mathbb{R}^{N \times M \times d}$ that unprojects the 2D features from each pixel to be 3D view-features at the corresponding Voint. This is performed by using the mapping $\mathbf{B}$ created by the renderer to form the Voint cloud features $\widehat{\mathcal{X}}$. Note that the points are not necessarily visible from all the views, and some Voints that are not visible from any of the $M$ views will not receive any features. We post-process these empty points ($\sim 0.5\%$ of points during inference) to be filled with nearest 3D neighbors features. The output Voint cloud features would be described as follows.

$$\begin{aligned} \widehat{\mathbf{x}}_i &= \left\{ \mathbf{g}_{i,j,:} \in \mathbb{R}^d \ \mid \ \mathbf{x}_i \in \mathcal{X} \ , \ \mathbf{V}_{i,j} = 1 \right\}_{j=1}^{M} \\ \mathbf{g}_{:,j} &= \Phi_{\mathbf{B}} \left( \mathbf{C}\left(\mathbf{R}\left(\mathcal{X}, \mathbf{u}_j\right)\right), \mathbf{B} \right) \\ \widehat{\mathcal{X}} &= \left\{ \widehat{\mathbf{x}}_i \in \mathbb{R}^{M \times d} \right\}_{i=1}^{N} \end{aligned} \tag{12}$$

## A.4 Voint Operations

**VointMax.** In order to learn a neural network in the Voint space in the form dictated by Theorem 1, we need to define some basic differentiable operations on the Voint space. The

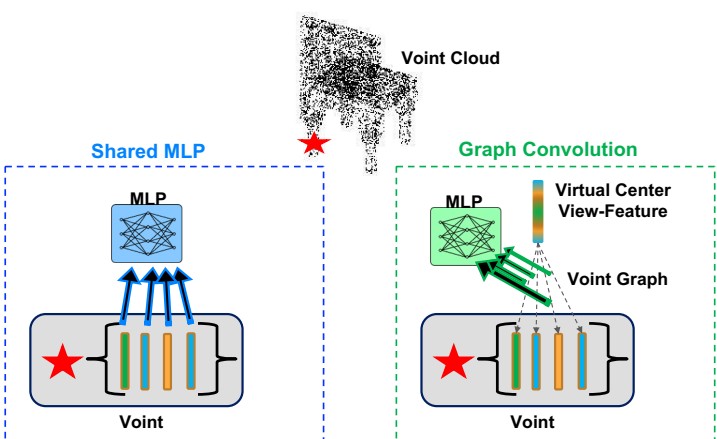

Figure 6: **VointNet Variants.** We propose three variants of VointNet that use three different examples of VointConv operation $h_v$: shared MLP (MLP), Graph Convolution (GCN), and Graph Attention (GAT). Here we highlight the main difference between VointNet (MLP) that shares the MLP on all the view-features and VointNet (GCN) that creates a fully connected graph on the view-features and learn an MLP on the edge view-features. VointNet (GAT) is similar to VointNet (GCN) in addition to learning attention weights for each view-feature in weighted average aggregation.

max operation on the Voint cloud can be defined as follows.

$$
\begin{aligned}
\text{VointMax}(\widehat{\mathbf{x}}) &= \max_{j} \widehat{\mathbf{x}}_{i,j}, \; \forall i, j \\
\text{s.t.} \;\; i &\in 1, 2, ..., N \; , \; j \in 1, 2, ..., M \;\; , \mathbf{V}_{i,j} = 1
\end{aligned}
\tag{13}
$$

Equivalently, $\text{VointMax}(\widehat{\mathbf{x}}) = \max_j \left( \widehat{\mathbf{x}}_{:,j} - \infty \overline{\mathbf{V}}_{:,j} \right)$, where $\overline{\mathbf{V}}$ is the complement of $\mathbf{V}$.

**VointConv.** We define the convolution operation $h_{\mathrm{V}} : \; \mathbb{R}^{N \times M \times d} \to \mathbb{R}^{N \times M \times d'}$ as any learnable function that operates on the Voint space with shared weights on all the Voints and has the view-features input size $d$ and outputs view-features of size $d'$ and consists of $l_V$ layers. Examples of this VointConv operation include the following operations applied *only* on the visible view-features: a shared MLP, a graph convolution, and a graph attention. We detail these operations later in Section A.6, which result in different non-exhaustive variants of VointNet.

## A.5   Learning on 3D Voint Clouds

**VointNet.** Typical 3D point cloud classifiers with a feature max pooling layer work as in Eq (14), where $h_{\mathrm{mlp}}$ and $h_{\mathrm{Pconv}}$ are the MLP and point Convolutional ($1 \times 1$ or edge) layers, respectively. This produces a K-class classifier $\mathbf{F}$.

$$
\mathbf{F}(\mathcal{X}) = h_{\mathrm{mlp}}(\max_{\mathbf{x}_i \in \mathcal{X}} \{h_{\mathrm{Pconv}}(\mathbf{x}_i)\})
\tag{14}
$$

Here, $\mathbf{F} : \; \mathbb{R}^{N \times 3} \to \mathbb{R}^K$ produces the logits layer of the classifier with size $K$. On the other hand, the goal of the VointNet model is to get multi-view point cloud features that can be used after which by any point cloud processing pipeline. The VointNet module $\widehat{\mathbf{F}} : \mathbb{R}^{N \times M \times d} \to \mathbb{R}^{N \times d}$ as follows.

$$
\widehat{\mathbf{F}}(\widehat{\mathcal{X}}) = h_{\mathrm{P}} \left( \text{VointMax} \left( h_{\mathrm{V}}(\widehat{\mathcal{X}}) \right) \right),
\tag{15}
$$

## A.6   VointNet Variants

We define the convolution operation $h_{\mathrm{V}} : \; \mathbb{R}^{N \times M \times d} \to \mathbb{R}^{N \times M \times d'}$ in VointNet from Eq (15) as any learnable function that operates on the Voint space with shared weights on all the

Voints and has the view-features input size $d$ and outputs view-features of size $d'$ and consists of $l_V$ layers. Examples of this VointConv operation include the following:

**Shared MLP.** It is the most basic Voint neural network. For layer $l$, the features of Voint i at view j is updated as follows to layer $l + 1$

$$\mathbf{h}_{i,j}^{l+1} = \rho \left( \mathbf{h}_{i,j}^{l} \mathcal{W}_\rho \right), \ \forall i, j \tag{16}$$
$$\text{s.t.} \ \ i \in 1, 2, ..., N \ , \ j \in 1, 2, ..., M \ \ , \mathbf{V}_{i,j} = 1$$

where $\rho$ is the shared MLP with weights $\mathcal{W}_\rho$ followed by normalization and nonlinear function ( *e.g.* ReLU) applied on all Voints independently at the visible views features for each Voint. This formulation extends the shared MLP formulation for PointNet (Qi et al., 2017a) to make the MLP shared across the Voints and the views-features.

**Graph Convolution (GCN).** Just like how DGCNN (Wang et al., 2019c) extended PointNet (Qi et al., 2017a) by taking the neighborhood information and extract edge features, we extend the basic VointNet formulation in Eq (15). We define a fully connected graph for each Voint along the views dimension by creating a center virtual node connected to all the view features ( similar to the classification token in ViT (Dosovitskiy et al., 2021)). This *center* virtual view-feature would be assigned the index $j = 0$ and can be initilized with zeros as the "cls" token in ViT (Dosovitskiy et al., 2021). Then, Voint graph convolution operation can be defined as follows to update the activations from layer $l$ to $l + 1$

$$\mathbf{h}_{i,j}^{l+1} = \rho \left( \left( \max_k \psi \left( (\mathbf{h}_{i,j}^{l}, \mathbf{h}_{i,k}^{l}) \mathcal{W}_\psi \right) \right) \mathcal{W}_\rho \right) \tag{17}$$
$$\forall i, j, k \ \ \text{s.t.} \ \ i \in 1, 2, ..., N \ , \ j \in 0, 1, ..., M$$
$$k \in 0, 1, ..., M \ \ , k \neq j \ \ , \ \mathbf{V}_{i,j} = 1$$

where $\rho, \psi$ are two different shared MLPs as in Eq (16). The difference between VointNet (MLP) and VointNet (GCN) is highlighted in Figure 6.

**Graph Attention (GAT).** Similar to how Point Transformer (Zhao et al., 2020) extended the graph convolution by adding attention to DGCNN (Wang et al., 2019c), we extend the basic Voint GraphConv formulation in Eq (17). Voint graph attention operation can be defined as follows to update the activations from layer $l$ to $l + 1$

$$\mathbf{h}_{i,j}^{l+1} = \rho \left( \left( \sum_{k=0, k \neq j}^{M} \eta_k \psi \left( (\mathbf{h}_{i,j}^{l}, \mathbf{h}_{i,k}^{l}) \mathcal{W}_\psi \right) \right) \mathcal{W}_\rho \right) \tag{18}$$
$$\forall i, j \ \ \text{s.t.} \ \ i \in 1, 2, ..., N \ , \ j \in 0, 1, ..., M$$
$$\eta_k = \zeta \left( \mathbf{h}_{i,k}^{l} \mathcal{W}_\zeta \right) \ \ , \ \mathbf{V}_{i,j} = 1$$

where $\rho, \psi, \zeta$ are three different shared MLPs as in Eq (16), and $\eta_k$ are the learned attention weights for each neighbor view-feature.

# B    DETAILED EXPERIMENTAL SETUP

## B.1    DATASETS

**ScanObjectNN: 3D Point Cloud Classification.** We follow the literature (Goyal et al., 2021; Hamdi et al., 2021) on testing 3D classification in the challenging ScanObjectNN (Uy et al., 2019) point cloud dataset, since it includes background and considers occlusions. The dataset is composed of 2902 point clouds divided into 15 object categories. We use 2048 sampled points per object for Voint learning. We benchmark on its variants: Object only, Object with Background, and the Hardest perturbed variant (PB_T50_RS variant). Visualization is provided in Figure 7 of some of the renderings used in training the 2D backbone in our pipeline.

**ShapeNet Core55: 3D Shape Retrieval.** The shape retrieval challenge SHREC (Sfikas et al., 2017) uses ShapeNet Core55 is a subset of ShapeNet (Chang et al., 2015) for benchmarking. The dataset consists of 51,162 3D mesh objects labeled with 55 object classes. The

training, validation, and test sets consist of 35764, 5133, and 10265 shapes. We create a dataset of point clouds by sampling 5000 points from each mesh object as in MVTN (Hamdi et al., 2021).

**ShapeNet Parts: 3D Part Segmentation.** ShapeNet Parts is a subset of ShapeNet (Chang et al., 2015) that consists of 13,998 point cloud objects for train and 2,874 objects for the test from 16 categories and 50 parts. It is designed for the part segmentation task (Yi et al., 2016). Visualization is provided in Figure 10 of some of the renderings used in training the 2D backbone in our pipeline colored with the ground truth segmentation labels.

**ModelNet40: 3D Shape Classification Occlusion Robustness.** ModelNet40 (Wu et al., 2015) is composed of 12,311 3D objects (9,843/2,468 in training/testing) labelled with 40 object classes. We sample 2048 points clouds from the objects following previous works (Qi et al., 2017b; Zhao et al., 2020). Visualization is provided in Figure 8 of some of the renderings used in training the 2D backbone in our pipeline.

## B.2 Metrics

**Classification Accuracy.** The standard evaluation metric in 3D classification is accuracy. We report overall accuracy (percentage of correctly classified test samples) and average per-class accuracy (mean of all true class accuracies).

**Retrieval mAP.** Shape retrieval is evaluated by mean Average Precision (mAP) over test queries. For every query shape $\mathbf{S}_q$ from the test set, AP is defined as $AP = \frac{1}{\text{GTP}} \sum_n^N \frac{\mathbb{1}(\mathbf{S}_n)}{n}$, where $GTP$ is the number of ground truth positives, $N$ is the size of the ordered training set, and $\mathbb{1}(\mathbf{S}_n) = 1$ if the shape $\mathbf{S}_n$ is from the same class label of query $\mathbf{S}_q$. We average the retrieval AP over the test set to measure retrieval mAP.

**Segmentation mIoU.** Semantic Segmentation is evaluated by mean Intersection over Union (mIoU) over pixels or points. For every class label, measure the size of the intersection mask between the ground truth points of that label and the predicted points as that label. Then, divide by the size of the union mask of the same label to get IoU. This procedure is repeated over all the labels, and averaging the IoUs gives mIoU. We report two types of mIoUs: Instance-averaged mIoU (averages all mIoUs across all objects ) and Category-averaged mIoU (averages all mIoU from shapes of the same category, and then average those across object categories).

## B.3 Baselines

**Point Cloud Networks.** We include PointNet (Qi et al., 2017a), PointNet++ (Qi et al., 2017b), DGCNN (Wang et al., 2019c), PVNet (You et al., 2018), and KPConv (Thomas et al., 2019), Point Transformer (Zhao et al., 2020) and CurveNet (Xiang et al., 2021) as baselines that use point clouds. These methods leverage different convolution operators on point clouds by aggregating local and global point information.

**Multi-View Networks.** We also compare against multi-view classification approaches like MVCNN (Su et al., 2015) and MVTN (Hamdi et al., 2021) as baselines for classification and retrieval. Since there is no available multi-view pipeline for 3D part segmentation, we adopt some of the multi-view segmentation baselines (*e.g.* Label Fusion (Wang et al., 2019a) and Mean Fusion (Kundu et al., 2020)) for part segmentation to work in the Voint space.

## B.4 Implementation Details

**Rendering and Un-Projection.** We choose the differentiable point cloud renderer $\mathbf{R}$ from Pytorch3D (Ravi et al., 2020) in our pipeline for its speed and compatibility with Pytorch libraries (Paszke et al., 2017). We render multi-view images with size $224 \times 224 \times 3$. We color the points by their normals' values or keep them white if the normals are not available. Following a similar procedure to (Wei et al., 2020; Hamdi et al., 2021), the view-point setup is randomized during training (using $M = 8$ views) and fixed to spherical views in testing (using $M = 12$ views).

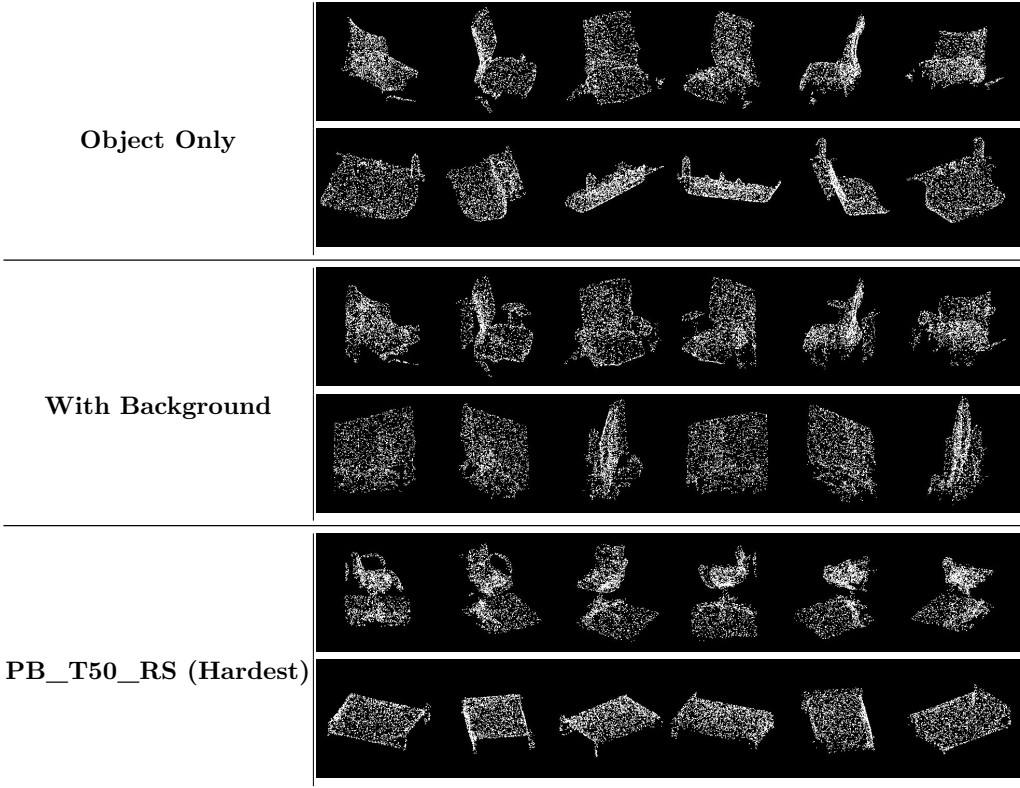

Figure 7: **ScanObjectNN Variants.** We show examples of point cloud renderings of different variants of the ScanObjectNN (Uy et al., 2019). These renderings are used in training VointNet for 3D point cloud classification.

**Architectures.** For the 2D backbone, we use ViT (Dosovitskiy et al., 2021) (with pretrained weights from TIMM library (Wightman, 2019)) for classification and DeepLabV3 (Chen et al., 2018) for segmentation. We used parallel heads for each object category for part segmentation since the task is solely focused on parts. We use the 3D cross-entropy loss on the 3D point cloud output and the 2D cross-entropy loss when the loss is defined on the pixels. When used, the linear tradeoff coefficient of the 2D loss term is set to 0.003. To balance the frequency of objects in part segmentation, we multiply the loss by the frequency of the object class of each object we segment. The feature dimension of the VointNet architectures is $d = 64$, and the depth is $l_V = 4$ layers in $h_V$. The main results are based on VointNet (MLP) variant unless otherwise specified. The coordinates $\mathbf{x}$ can be optionally appended to the input view-features $\widehat{\mathbf{x}}$, which can improve the performance but reduce the rotation robustness as we show later in Section C.1 and Table 9.

**Training Setup.** We train our pipeline in two stages, where we start by training the 2D backbone on the 2D projected labels of the points, then train the full pipeline end-to-end while focusing the training on the VointNet part. We use the AdamW optimizer (Loshchilov & Hutter, 2017) with an initial learning rate of 0.0005 and a step learning rate schedule of 33.3% every 12 epochs for 40 epochs. The pipeline is trained with *one* NVIDIA Tesla V100 GPU. We do not use any data augmentation.

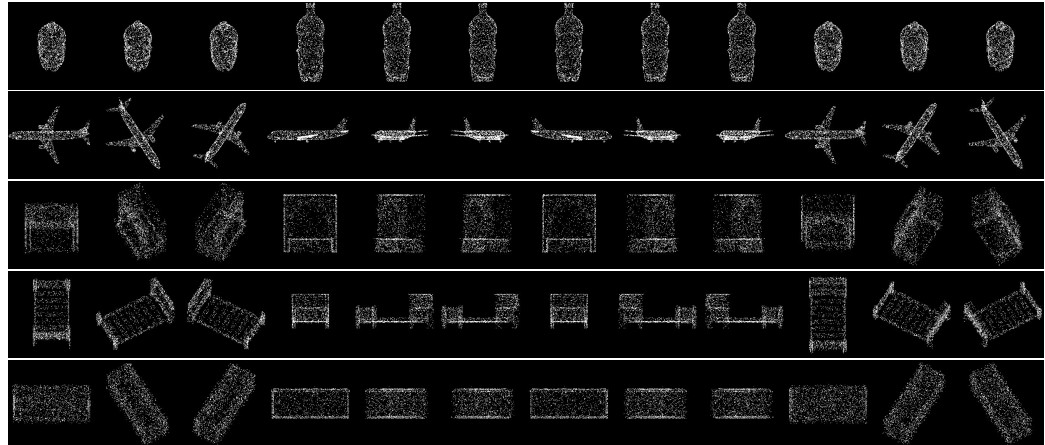

Figure 8: **ModelNet40.** We show some examples of point cloud renderings of ModelNet40 (Wu et al., 2015) used for 3D classification robustness in our setup.

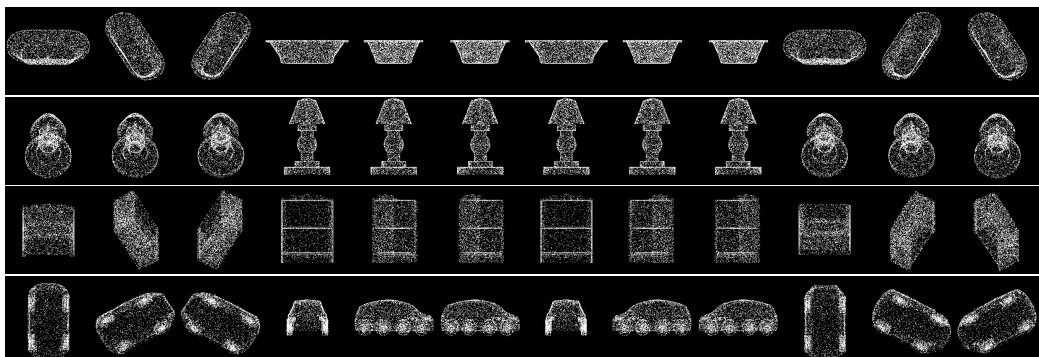

Figure 9: **ShapeNet Core55.** We show some examples of point cloud renderings of ShapeNet Core55 (Chang et al., 2015) used for 3D shape retrieval in our setup.

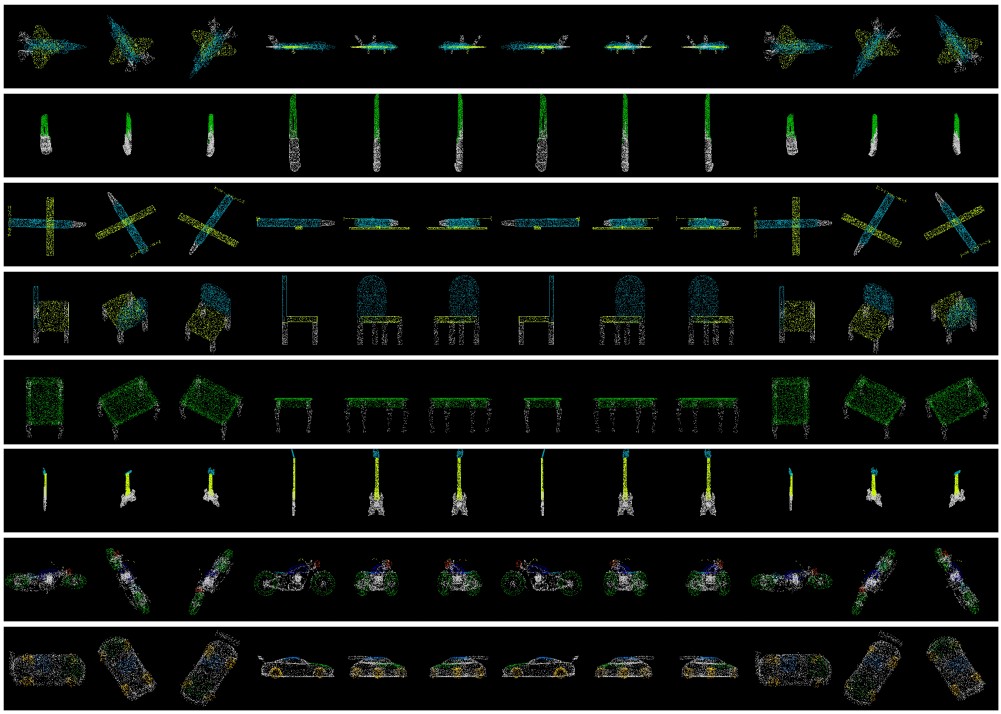

Figure 10: **ShapeNet Parts.** We show some examples of point cloud renderings of ShapeNet Parts (Yi et al., 2016) colored with ground truth segmentation labels. We use these renderings as 2D ground truth to pre-train the 2D backbone **C** for 2D segmentation before training VointNet's pipeline for 3D segmentation.

| Method | Data Type | Classification ModelNet40 | Shape Retrieval ShapeNet Core |
|---|---|---|---|
| PointNet (Qi et al., 2017a) | Points | 89.2 | - |
| PointNet++ (Qi et al., 2017b) | Points | 91.9 | - |
| DGCNN (Wang et al., 2019c) | Points | 92.2 | - |
| KPConv(Thomas et al., 2019) | Points | 92.9 | - |
| PCT(Guo et al., 2021) | Points | 93.3 | - |
| CurveNet(Xiang et al., 2021) | Points | **93.8** | - |
| ReVGG (Sfikas et al., 2017) | M-View | - | 74.9 |
| MVCNN (Su et al., 2015) | M-View | 90.1 | 73.5 |
| ViewGCN (Wei et al., 2020) | M-View | 93.3 | 78.4 |
| MVTN (Hamdi et al., 2021) | M-View | **93.8** | 82.9 |
| VointNet (ours) | Voints | 92.8 | **83.3** |

Table 7: **3D Shape Classification and Retrieval**. We report VointNet's classification accuracy on ModelNet40 (Wu et al., 2015) and its 3D shape retrieval mAP on ShapeNet Core55 (Chang et al., 2015; Sfikas et al., 2017). Baseline results are reported from (Hamdi et al., 2021; Zhao et al., 2020; Xiang et al., 2021).

| Method | Rotation Perturbations Range | | |
|---|---|---|---|
| | 0° | ±90° | ±180° |
| PointNet (Qi et al., 2017a) | 88.7 | 42.5 | 38.6 |
| PointNet ++ (Qi et al., 2017b) | 88.2 | 47.9 | 39.7 |
| RSCNN (Liu et al., 2019a) | 90.3 | 90.3 | 90.3 |
| MVTN (Hamdi et al., 2021) | 91.7 | 90.8 | **91.2** |
| VointNet (ours) | 91.5 | **90.9** | 91.1 |

Table 8: **Rotation Robustness for 3D Classification.** At test time, we randomly rotate objects in ModelNet40 (Wu et al., 2015) around the Y-axis (gravity) with different ranges and report the overall accuracy.

## C  ADDITIONAL RESULTS

### C.1  MODEL ROBUSTNESS

**Rotation Robustness for 3D Classification.** We follow the standard practice in 3D shape classification literature by testing the robustness of trained models to perturbations at test time (Liu et al., 2019a; Hamdi et al., 2021). We perturb the shapes with random rotations around the Y-axis (gravity-axis) contained within ±90° and ±180° and report the test accuracy over ten runs in Table 8.

**Rotation Robustness for 3D Segmentation.** We follow the previous 3D literature by testing the robustness of trained models to perturbations at test time (Liu et al., 2019a; Hamdi et al., 2021; 2020). We perturb the shapes in ShapeNet Parts with random rotations in $SO(3)$ at test time (ten runs) and report Ins. mIoU in Table 9. Note how our VointNet performance largely exceeds the baselines in this realistic unaligned scenario. We can augment the training with rotated objects for the baselines, which improves their robustness, but loses performance on the unrated setup. Adding $xyz$ coordinates to the view-features of VointNet improves the performance on an unrotated setup but negatively affects the robustness to rotations. The discrepancy between the Voint results and the results of some point cloud methods is that Voints heavily depend on the underlying 2D backbone and inherit all its biases, especially those from pretraining. Hence, the 2D backbone limits what the performance can reach with VointNet. We study the effect of the backbone in detail in Section C.2. Figure 11 shows qualitative 3D segmentation results for VointNet and Mean Fuse (Kundu et al., 2020) as compared to the ground truth.

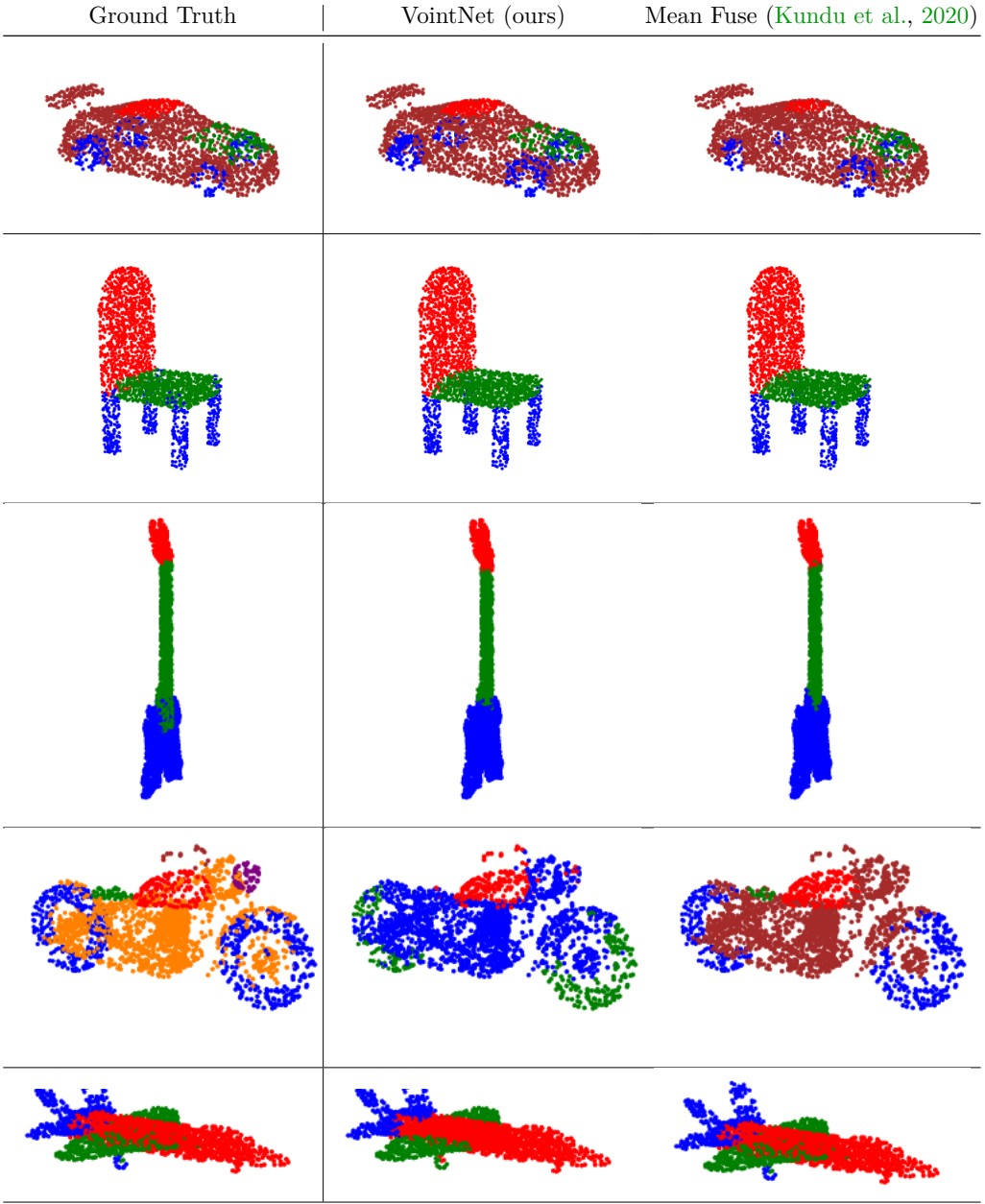

Figure 11: **Qualitative Comparison for 3D Part Segmentation.** We compare our VointNet 3D segmentation prediction to Mean Fuse (Kundu et al., 2020) that is using the same trained 2D backbone. Note how VointNet distinguishes detailed parts (*e.g.* the car window frame). Beware that visualization colors can shift if an extra label is predicted (*e.g.* the motorbike labels are correct).

| Method | Segmentation Under Rotation | |
| | Unrotated | Rotated |
| --- | --- | --- |
| PointNet (Qi et al., 2017a) | 80.1 | 36.6 ±0.2 |
| DGCNN (Wang et al., 2019c) | 80.1 | 37.1 ±0.2 |
| PointNet + Aug. | 65.8 | 65.8 ±0.1 |
| DGCNN + Aug. | 60.7 | 60.7 ±0.2 |
| Mean Fuse (Kundu et al., 2020) | 79.1 | 61.6 ±0.1 |
| Label Fuse (Wang et al., 2019a) | 78.9 | 61.0 ± 0.1 |
| VointNet (w/o $xyz$) | 79.6 | 65.4 ±0.1 |
| VointNet (w/o $xyz$) + Aug. | 68.0 | **68.5** ± 0.1 |
| VointNet (w/ $xyz$) | **81.2** | 61.5 ±0.2 |

Table 9: **Rotation Robustness for 3D Part Segmentation.** At test time, we randomly rotate objects from ShapeNet Parts (Yi et al., 2016) and report the Ins. mIoUs of our VointNet compared to trained PointNet (Qi et al., 2017a) and DGCNN (Wang et al., 2019c). Note how VointNet's performance largely exceeds the baselines in realistic unaligned scenarios, highlighting the benefit of view dependency. If we use rotation augmentation in training for the baselines, the rotated performance improves, but the unrotated performance drops.

## C.2 DETAILED ANALYSIS

**Effect of Pretraining.** We study the effect of pretraining the 2D backbone **C** for 3D classification on ModelNet40. Training a ViT with Mean Fuse for 3D classification on ModelNet40 obtains 92.2 test Acc. with ImageNet pretraining and 80.0 test Acc. from scratch. Other multi-view networks, *e.g.* MVCNN (Su et al., 2015), ViewGCN(Wei et al., 2020), and MVTN(Hamdi et al., 2021) all use ImageNet pretraining, which is not unique to Voints.

**Classification Backbone.** We study the effect of ablating the 2D backbone **C** for 3D classification on ModelNet40. We show in Table 10 the performance of VointNet (MLP) when Vit-B (Dosovitskiy et al., 2021) and ResNet-18 (He et al., 2015) are used. We also show that following the per-point classification setup instead of the per-shape for 3D shape classification leads to worse performance for VointNet and the naive multi-view. This is why we used the per-shape approach when adopting VointNet for 3D classification (using one Voint for the entire shape).

**Number of points and visibility.** Table 11 studies the effect of point number on 3D part segmentation performance, when different numbers of views are used. The visibility ratio is also reported in each case.

**Points color.** We colored the points with ground truth normals as in Figure 16, when they are available (ShapeNet Parts), and we used white colors as in Figure 9, when other baselines do not use normals. We ablate the color of the points on VointNet (MLP) with normals colors, white color, and NOCs colors (Wang et al., 2019b). We obtain the following segmentation mIoU results: (*normals*: 80.6), (*white*: 74.7), and (*NOCs*: 57.9).

**Time and Memory Requirements.** To assess the contribution of the Voint module, we take a macroscopic look at the time and memory requirements of each component in the pipeline. We record the number of floating-point operations (GFLOPs) and the time of a forward pass for a single input sample. In Table 12, the VointNet module contributes negligibly to the memory requirements compared to multi-view and point networks.

**Feature Size ($d$).** We study the effect of the feature size $d$ on the performance of VointNet (MLP) in 3D part segmentation on ShapeNet Parts (Yi et al., 2016) and plot the results ( with confidence intervals) in Figure 12. We note that the performance peaks at $d = 128$, but it is close to what we use in the main results ($d = 64$).

| View Aggregation | 2D Backbone | | |
|:---:|:---:|:---:|:---:|
| | ResNet18 (per-shape) | ViT-B (per-shape) | DeepLabV3 (per-point) |
| VointNet | 91.2 | **92.8** | 10.2 |

Table 10: **Ablation Study for 3D Classification**. We study the effect of different 2D backbone for ModelNet40 3D classification task. We compare VointNet's performance to naive multi-view (*e.g.* MVCNN (Su et al., 2015) or Mean Fuse (Kundu et al., 2020)) using the same 2D backbone. Note that using the per-point classification setup instead of the per-shape for 3D shape classification leads to worse performance for VointNet and the naive multi-view.

| Points # | Metric | Number of Views | | | |
|:---:|:---:|:---:|:---:|:---:|:---:|
| | | 2 | 4 | 8 | 12 |
| 500 | visibility | 99.1 | 99.9 | 100 | 100 |
| | mIoU | 69.2 | 73.9 | 76.0 | 76.4 |
| 1000 | visibility | 98.0 | 99.7 | 100 | 100 |
| | mIoU | 69.5 | 74.3 | 76.5 | 77.1 |
| 2000 | visibility | 95.7 | 99.2 | 99.8 | 99.9 |
| | mIoU | 69.7 | 75.0 | 77.7 | 78.5 |

Table 11: **Analysis on Number of Points and Visibility**. We show the Instance mIoUs and visibility ratio $(1 - \frac{\text{empty}}{\text{total}})\%$ of our VointNet on ShapeNet Parts when varying points # and number of views.

**Model Depth ($l_v$).** We study the effect of the model depth $l_v$ on the performance of VointNet (MLP) in 3D part segmentation on ShapeNet Parts (Yi et al., 2016) and plot the results ( with confidence intervals) in Figure 13. We note that model depth of VointNet does not enhance the performance significantly. Our choice of $l_v = 4$ balances the performance and the memory/computations requirements of VointNet (MLP).

**Distance to the Object.** We study the effect of distance to the object in rendering as in Figure 17 to the performance of VointNet (MLP) in 3D part segmentation on ShapeNet Parts (Yi et al., 2016) and plot the results ( with confidence intervals) in Figure 14. We note that our default choice of 1.0 is actually reasonable. This choice of distance shows the object entirely ( as illustrated in Figure 17), but also cover the details needed for small parts segmentation (see Figure 11).

**Image Size ($H, W$).** We study the effect of the image size $H\&W$ on the performance of Mean Fuse (Kundu et al., 2020) baseline when training the 2D backbone for 3D part segmentation. We plot the results ( with confidence intervals) in Figure 15.

**Number of Views on Classification.** We study the effect of the number of views (M) on classification accuracy on ModelNet40 Wu et al. (2015) of VointNet and report results in Table 13.

**Unprojection Operation Speed.** We evaluate the speed of the unprojection operation $\mathbf{\Phi_B}$ and report average latency of 10,000 runs (in ms) in Table 14.

**Unprojection Operation Speed.** We evaluate the speed of the point cloud renderer $\mathbb{R}$ used in Voint pipeline from Pytroch3D Ravi et al. (2020) and report average latency of 1,000 renderings (in ms/image) in Table 15.

## C.3 VISUALIZATIONS

In Figure 16 and 17, we visualize the multi-view renderings of the point clouds along with the 2D learned features based on the DeepLabV3 (Chen et al., 2018) backbone. These features are then unprojected and transformed by VointNet to obtain 3D semantic labels.

| Network | GFLOPs | Time (ms) | Parameters # (M) |
|---|---|---|---|
| MVCNN (Su et al., 2015) | 43.72 | 39.89 | 11.20 |
| ViewGCN (Wei et al., 2020) | 44.19 | 26.06 | 23.56 |
| ResNet 18 (He et al., 2015) | 3.64 | 3.70 | 11.20 |
| ResNet 50 (He et al., 2015) | 8.24 | 9.42 | 23.59 |
| ViT-B (Dosovitskiy et al., 2021) | 33.70 | 12.46 | 86.57 |
| ViT-L (Dosovitskiy et al., 2021) | 119.30 | 29.28 | 304.33 |
| FCN (Long et al., 2015) | 53.13 | 10.34 | 32.97 |
| DeeplabV3 (Chen et al., 2018) | 92.61 | 20.62 | 58.64 |
| PointNet (Qi et al., 2017a) | 1.78 | 4.24 | 3.50 |
| DGCNN (Wang et al., 2019c) | 10.42 | 0.95 | 16.350 |
| MVTN (Hamdi et al., 2021) | 1.78 | 4.24 | 3.5 |
| VointNet (MLP) | 1.90 | 2.90 | 0.04 |
| VointNet (GCN) | 16.18 | 32.10 | 0.05 |
| VointNet (GAT) | 32.05 | 68.71 | 0.07 |
| Full Voint pipeline | 94.51 | 23.50 | 58.68 |

Table 12: **Time and Memory Requirements**. We assess the contribution of the Voint module to the time and memory requirements in the multi-view and point cloud pipeline. Note that VointNet (shared MLP) is almost 100 times smaller than PointNet (Qi et al., 2017a).

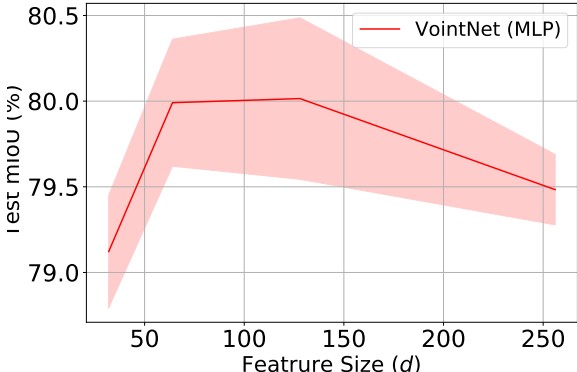

Figure 12: **The Effect of Feature Size** $d$**.** We plot Ins. mIoU of 3D segmentation *vs.* the feature size $d$ used in training on ShapeNet Parts (Yi et al., 2016). We note that the performance peaks at $d = 128$, but it is close to what we use in the main results ($d = 64$).

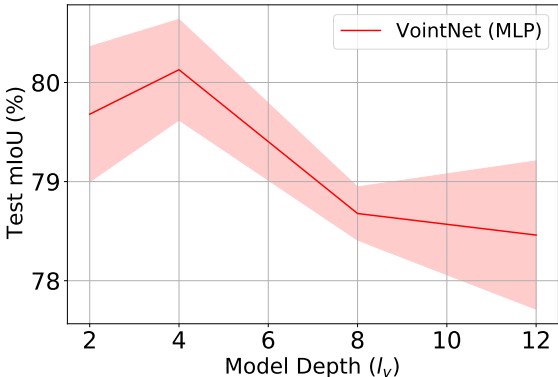

Figure 13: **The Effect of Model Depth $l_v$.** We plot Ins. mIoU of 3D segmentation *vs.* the model depth $l_v$ used in training on ShapeNet Parts (Yi et al., 2016). We note that model depth of VointNet does not enhance the performance significantly. Our choice of $l_v = 4$ balances the performance and the memory/computations requirements of VointNet (MLP).

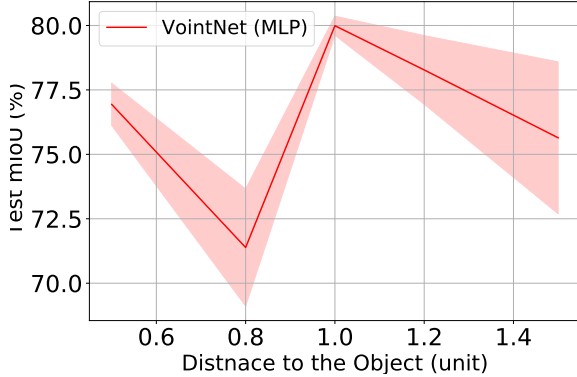

Figure 14: **The Effect of Distance to the Object.** We plot Ins. mIoU of 3D segmentation *vs.* the distance to the object used in inference on ShapeNet Parts (Yi et al., 2016). We note that our default choice of 1.0 is actually reasonable. This choice of distance shows the object entirely ( as illustrated in Figure 17), but also cover the details needed for small parts segmentation (see Figure 11).

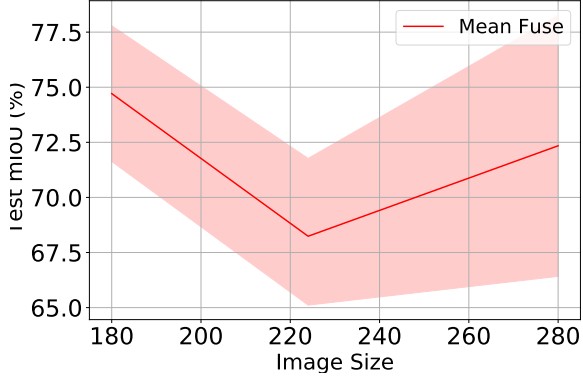

Figure 15: **The Effect of Image Size $H, W$.** We plot Ins. mIoU of 3D segmentation *vs.* the image size used in inference on ShapeNet Parts (Yi et al., 2016).

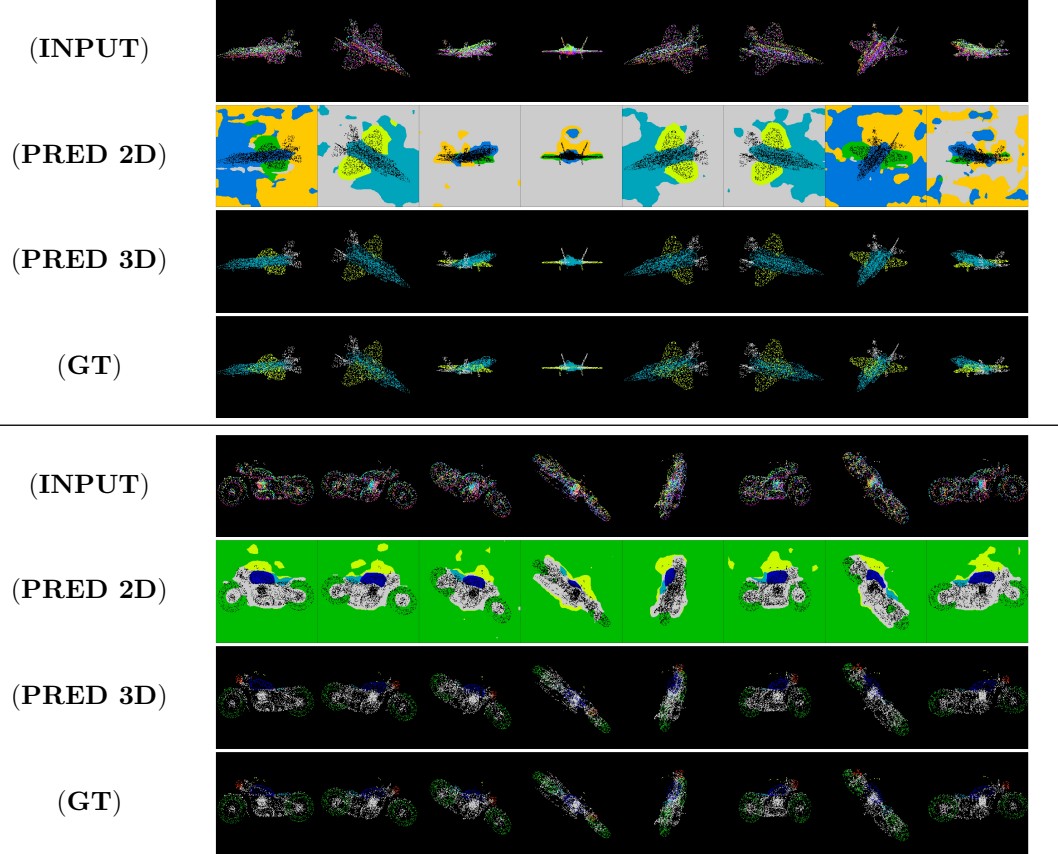

Figure 16: **Multi-view Projected Segmentation 1.** We show how, after rendering points, we can segment in the image space. For each example, we show (*INPUT*): the projections of the points (colored with normals) used in training with random view-points. (*PRED 2D*): the segmentation prediction of the 2D backbone (DeepLabV3) (Chen et al., 2018)). (*PRED 3D*): the unprojected 3D segmentation prediction. (*GT*): the 3D segmentation ground truth.

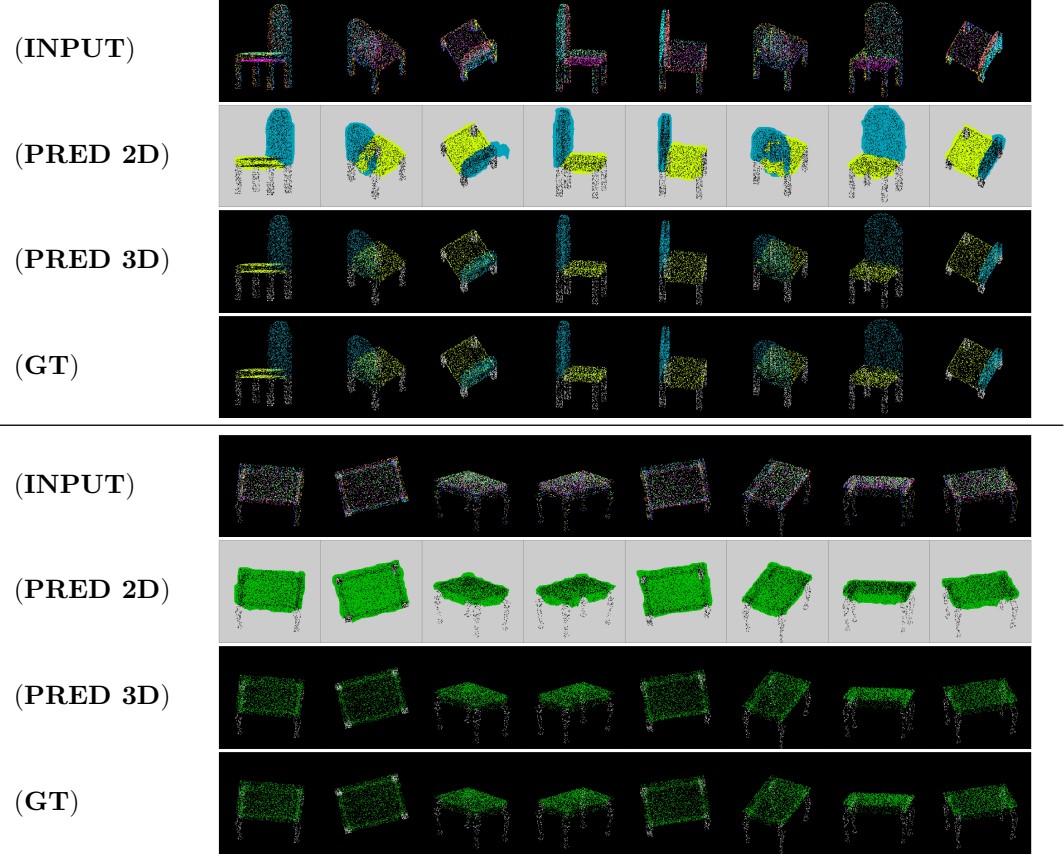

Figure 17: **Multi-view Projected Segmentation 2.** We show how, after rendering points, we can segment in the image space. For each example, we show (*INPUT*): the projections of the points (colored with normals) used in training with random view-points. (*PRED 2D*): the segmentation prediction of the 2D backbone (DeepLabV3) (Chen et al., 2018)). (*PRED 3D*): the unprojected 3D segmentation prediction. (*GT*): the 3D segmentation ground truth.

| Method | Number of Views | | | |
|---|---|---|---|---|
| | 4 | 6 | 8 | 10 |
| VointNet (Cls. Acc.) | 90.3 | 90.8 | 92.0 | 92.3 |

Table 13: **Effect of the Number of Views on Classification**. We report the classification accuracy of VointNet vs. the number of views (M) used in the training on ModelNet40.

| Method | Number of Views | | | | | | |
|---|---|---|---|---|---|---|---|
| | 1 | 2 | 4 | 6 | 8 | 10 | 12 |
| Features Unprojection | 3.0 | 5.3 | 11.45 | 15.7 | 17.2 | 29.7 | 24.0 |
| Labels Unprojection | 2.6 | 2.5 | 3.4 | 3.1 | 3.0 | 3.2 | 3.6 |

Table 14: **Unprojection Operation Speed.** We report the average latency (in ms) over 10,000 runs of the unprojection operation with its two forms: features unprojection (used in mean) and labels unprojection (used in mode).

| Criteria | Number of Points | | | | |
|---|---|---|---|---|---|
| | $1e2$ | $1e3$ | $1e4$ | $1e5$ | $1e6$ |
| Point Rendering Speed (ms/image) | 7.2 | 7.6 | 7.7 | 10.4 | 37.7 |

Table 15: **Point Rendering Speed.** We report the average rendering speed (in ms/image) over 1,000 renderings of the point cloud renderer Ravi et al. (2020) used in Voint clouds.

