# OpenReview forum: "Voint Cloud: Multi-View Point Cloud Representation for 3D Understanding "
_ICLR.cc/2023/Conference — ICLR 2023 poster_

### Official Review · Reviewer_aJdJ · 2022-10-24

**Confidence:** 5
**Correctness:** 4
**Technical Novelty And Significance:** 2
**Empirical Novelty And Significance:** 2
**Recommendation:** 8

**Clarity, Quality, Novelty And Reproducibility:**

The paper is well written and all ideas are well explained
The novelty of the paper is limited but the method is well executed
The authors provide their code for reproducibility

**Strength And Weaknesses:**

The paper is well written. The approach is well evaluated and the results are strong. The proposed idea is a well-executed derivation of existing art.

Weaknesses:

1) The novelty of the paper is limited. Essentially, the authors create am image based viewing track of a 3D point (common in SFM and SLAM) and modify pointNet to work in a higher dimensionality space.

2) There is too much effort from the authors to establish the term Voint (It is not clear if there is a fundamental difference between max pooling and convolution with VointMax and VointConv other than applying the operation on higher dimension. )

3) The method is assuming that the render is able to project 3D points to 2D with perfect occlusion detection. Have the authors considered how VointCloud can be extended to more complex point-clouds where mesh reconstruction from a renderer may not guarantee high occlusion detection accuracy?

4) How efficient is VointCloud and VointNet in terms of computational and space complexity?
Minor comments
1) Un-projection: The correct term is back-projection.
2) Eq2 M,N are undifined symbols
3) Table 3 ShapeNet

**Summary Of The Paper:**

The authors propose the use of image based features coupled with 3D points to train a network to perform various 3D perception tasks, such as classification, retrieval or segmentation, on point clouds.  The authors demonstrate state of the art performance on synthetic several datasets.

**Summary Of The Review:**

Although I have some concerns about the overall terminology and limited novelty of the paper, I think that the method is well executed and the results support the direction. I recommend the paper for publication.

---

> ### Author Response · Authors · 2022-11-18
> **rebuttal 1**
>
> ## P1. Novelty
> We respectfully disagree with the reviewer and agree with the other reviewers, who state that the main idea is novel. The novelty is _not limited_ to the point rendering and VointNet. The methodology includes the means of  extracting multi-view deep features (_e.g._ from 2D backbones) and aggregating view-features _per-point_ for 3D understanding tasks. Also, the proposed modular end-to-end deep learning pipeline for points and multi-view 2D backbone for 3D classification/segmentation is an important contribution of this work to the community.
>
> ## P2. Voint Operations
> There is one important distinction between Voint operations and point operations other than the higher dimension. The visibility of the rendered points $\mathbf{V}$ plays a role in the definitions of the Voint operations as can be seen in Equations 13, 15 and 16 in the Appendix. This is important because not all $N$ points are seen from all $M$ views, and the visibility provided by the renderer $\mathbf{R}$ controls Voint learning. As a result, VointMax and VointConv are _not_ simply a max pooling and a convolution operation on higher dimensions.
>
> ## P3. Occlusion and Visibility of Rendered Points
> We appreciate the concern of the reviewer. However, our rendering module does not need a mesh reconstruction to distinguish foreground from background points as it renders them all, with priority given to the foreground points if two points were projected onto the same pixel. This makes segmenting 3D points more difficult by a 2D network from a single view. However, as the number of views increases, the capability of VointNet to distinguish and aggregate the 2D features for _all_ points increases (Figure 4). Rendering all points is important because it guarantees visibility.
> Usually, if the number of points in the 3D point cloud to be analyzed is not very large, the visibility of the rendered points ($1-\frac{\text{empty}}{\text{total}}$) is almost 100\% . We provide an analysis of visibility _vs._ the number of points in Table 11 of Appendix Section C.2.
>
> ## P4. Latency and Cost Analysis
> We provide a thorough cost analysis of different components of the Voint pipeline in Table 12 in Appendix C.2.
>
> ## P5. Unprojection Terminology
> We thank the reviewer for the suggestion and will refer to the unprojection operation as back-projection instead.
>
> ## P6. M, N
> $M$: Number of views, and, $N$: Number of points, as defined in Section 3 and highlighted in Figure 2.
>
> ## P7. Spelling
> We thank the reviewer for the suggestion. We fixed it in the revised version.

---

### Official Review · Reviewer_14aj · 2022-10-25

**Confidence:** 4
**Correctness:** 4
**Technical Novelty And Significance:** 3
**Empirical Novelty And Significance:** 3
**Recommendation:** 6

**Clarity, Quality, Novelty And Reproducibility:**

The paper is overall clear and well-written, with some typos to fix. The idea is interesting and novel. The authors have provided their code.

**Strength And Weaknesses:**

Strength
- Though counter-intuitive, the idea of extracting features in 2D and fusing in 3D is interesting, clean, and neat. The method achieves very impressive performance on all tasks while keeping a compact model size and fast inference efficiency.
- Experiments have shown that the proposed backbone is less prone to partial observation and change of orientation.

Weaknesses
- The authors have not studied how noise will affect the backbone performance. It could be much harder to extract useful information in 2D than in 3D for noisy point clouds. This could potentially be a case where 3D methods take advantage.
- The authors seem to deliberately use different view numbers (8 vs. 12) and sampling methods (random vs. spherical) during training and testing. This is not explained and studied in the paper.
- The authors should include more details about the renderer e.g. speed, color?, point size, etc.


- If it is possible to render point clouds from different distances (say 0.5, 1.0, and 2.0) and use them as multi-scale features? Many 2D and 3D backbones employ similar ideas explicitly. Any discussion will be appreciated.

Minor
- Table 3: SahpeNet -> ShapeNet
- Figure 14: Distnace -> Distance
- Figure 12-15 appear to be very large and not nicely rendered (14,15)

**Summary Of The Paper:**

The paper proposed a novel point cloud backbone. The backbone renders point clouds into 2D multi-view renderings and extracts 2D features with 2D backbones. These features are then unprojected back onto 3D point clouds where they are fused into final point features.
The backbone is very compact and efficient.
The authors have performed extensive experiments including 3D point cloud classification, shape retrieval, and 3D part segmentation, and shown good performance on all benchmarks.

**Summary Of The Review:**

I'm leaning toward acceptance due to the impressive performance and efficiency of the method. Adding more discussions and completing the details suggested above could further strengthen my recommendation.

---

> ### Author Response · Authors · 2022-11-18
> **rebuttal 1**
>
> ## P1. Performance vs Noise Level
> We respectfully disagree with the reviewer that point-based methods are better with increasing noise levels. In fact, we believe it is the opposite. Table 2 of the noisy and realistic ScanObjectNN dataset highlights this. The benchmark has three variants: Object-Only (clean), Object-with-Background (some noise), and Hardest (high noise levels that include cropping and rotations). As can be seen, the performance of point-based methods (like PointNet++ and DGCNN) drop drastically with the increasing noise level, while multi-view methods (MVTN and Voints) maintain a relatively stable performance even in the case of  extreme noise. Please see Figure 7 for examples of this dataset compared to the clean ModelNet40 and ShapeNet in Figures 8 and 9 in the Appendix.
>
> ## P2. Number of Views at Test Time
> One advantage of multi-view methods is the correlation between performance and the number of views (as observed in Figure 4). However, the cost of training linearly increases with the number of views, which can slow down training when a large number of views is considered. So, we opt for using 8 views in training, while increasing views at test time to improve test performance for ``free'', _i.e._ without the need for training on the large-number-of-views setup.
> During training, the views are randomly sampled, which can be considered as a way to augment the training set to accommodate any views.
>
> ## P3. Renderer Information
> The point radius is fixed at 0.006 and the image resolution is 224 $\times$ 224. All other essential configurations of our Pytorch3D renderer (including the colors) are studied in detail in Appendix Section C.2. The remaining configurations can be found in the attached code in the files: _config.yaml_ and _run_voints.py_ CLI.
> The speed analysis of the renderer can be found below and is added as Table 15 to the updated Appendix Section C.2 as well.
>
> number of points: 1e2 - 1e3 - 1e4 - 1e5 - 1e6
>
> -----------------------------------------------------------------
>
> rendering speed (ms/image) : 7.2 , 7.6 ,7.7 , 10.4 , 37.7
>
> ## P4. Multi-Scale Suggestion
> We thank the reviewer for the suggestion. We agree that using multiple scales is a good idea to leverage the 2D features at different scales, which capture different aspects of the 3D scene. Tackling the engineering aspect of running multi-scale multi-view on multiple points ($M \times N \times S$) is an interesting future direction and would benefit the community.
>
> ## P5. Improving Figures 12-15
> We thank the reviewer for the suggestion. We updated the figures in the Appendix.

---

### Official Review · Reviewer_LsWo · 2022-10-26

**Confidence:** 3
**Correctness:** 4
**Technical Novelty And Significance:** 3
**Empirical Novelty And Significance:** 3
**Recommendation:** 6

**Clarity, Quality, Novelty And Reproducibility:**

The paper is well-written, idea is novel.

Typo: table 3 SahpeNet->ShapeNet

**Strength And Weaknesses:**

Strength:
- The idea of extracting low-level features from multi-view 2D images first and then aggregating back into 3D is novel. This takes advantage of mature 2D networks and still maintain the 3D geometry.
- The paper is well-written with extensive experiments and ablation studies.

Weakness:
I would like to see latency analysis of each component of proposed method and also in comparison with other methods. Looks like unprojection requires a lot of irregular memory access.

**Summary Of The Paper:**

This paper proposes a multi-view 3D point cloud representation called Voint cloud. The multi-view features are extracted from 2D networks and then aggregated into 3D point clouds using correspondence. Visibility of the points in each view is also attached. The proposed method reaches state-of-the-art-performance on several 3D understanding tasks, including 3D shape classification, retrieval, and robust part segmentation.

**Summary Of The Review:**

The idea is somewhat novel and results show the effectiveness of proposed method. Incremental change compared to previous method. It'd be nice to see the latency analysis.

---

> ### Author Response · Authors · 2022-11-18
> **rebuttal 1**
>
> ## P1. Latency and Cost Analysis
> We provide a thorough cost analysis of different components of the Voint pipeline in Table 12 in Appendix C.2. This assumes that a single view is used. Regarding the unprojection, the speed would depend on the number of views. For features and label unprojections of a single shape, the latency (in ms) on NVIDIA Titan GPU  for different numbers of views and with 2500 points is as follows. The results are averaged over 10,000 runs and the table is added in the revised Appendix Section C.2 as Table 14. We thank the reviewer for the suggestion. The views are fixed at test time for a stable and fair evaluation.
>
> number of views (M): 1 - 2 - 4 - 6 - 8 - 10 - 12
>
> -----------------------------------------------------------------
>
> features unprojection (mean) : 3.0 , 5.3 , 11.45 , 15.7 , 17.2 , 29.7, 24.0
>
> -----------------------------------------------------------------
>
> labels unprojection (mode):  2.6, 2.5, 3.4, 3.1, 3.0, 3.2, 3.6
>
> Note that the latency per view is negligible as compared to the 2D CNN forward pass, which constitutes the slowest module in the pipeline. However, with advancements in 2D computer vision, we expect further improvements in the voint cloud pipeline speed. Also, note that label unprojection is iterating over labels and hence less correlated to the number of views.

---

### Official Review · Reviewer_KkFB · 2022-10-31

**Confidence:** 5
**Correctness:** 4
**Technical Novelty And Significance:** 4
**Empirical Novelty And Significance:** Not applicable
**Recommendation:** 6

**Clarity, Quality, Novelty And Reproducibility:**

The paper is well-written for the most part. I would recommend incorporating one item or two from the appendix to the main paper.
It brings a new idea -- new 3D representation VointCloud and also has methodological novelty. The experiments seem to be reproducible.



**Strength And Weaknesses:**

Strengths:
+ New 3D representation which seamlessly integrates view-dependent representation for each 3D point
+ Proposed new pooling and convolution operation on the Voint Cloud leading to a new deep neural network called VointNet
+ state-of-the-art performance on 3D point cloud classification and 3D shape retrieval tasks using the proposed Voint Cloud representation

Weaknesses:
+ Generalizability to outdoor 3D scene understanding task: All experiments were done on indoor 3D scene understanding tasks (ScanObjectNN, ScanNet). To make a more generalizable claim for Voint Cloud representation, at least one experiment should have been done on an outdoor 3D scene understanding task, e.g., the KITTI dataset. Is there any specific reason for not doing that?


+ 3D part segmentation: The proposed method achieved better results against other baselines on a more challenging setup (rotated); however, CurveNet still retained the best performance 84.9 % mIoU on the "Unrotated setup." The paper did not discuss what could be the possible reason for this slight discrepancy.  The ablation study in Table 6 shows different backbones and different variations of VointNet. It is unclear which result from that table corresponds to the one reported in Table 4. I also acknowledge the fact that the paper mentions that unless otherwise stated, the default method for VointNet is shared MLP.

+ Backbone in 3D part segmentation: The backbones used in the 3D part segmentation task are widely used methods (FCN, DeepLabv3). The newer backbone might be more effective and possibly could improve the results. Please refer to the model "HRNet-W48" which was used in the following work: "MSeg: A Composite Dataset for Multi-domain Semantic Segmentation - Lambert, John, et al. CVPR'20"

+ Number of views: The authors showed an ablation study for the effect of the number of views on the performance of the Voint Cloud. However, it was shown only for the 3D part segmentation task. How about the 3D point cloud classification and 3D object retrieval tasks?

+ 3D point cloud classification and 3D shape retrieval experiment: The proposed Voint Cloud-based representation achieved state-of-the-art results. It would be interesting to show the comparison against methods with deformable convolution on Point Cloud, such as
"KPConv: Flexible and Deformable Convolution for Point Clouds - Hugues Thomas et al. ICCV'19"


+ VointNet variations: The paper shows three different variations of VointNet, the simplest of which is a shared MLP-based architecture. Figure 6 (in the appendix) could be included in the main paper to describe the method clearly.




**Summary Of The Paper:**

This paper presents a novel 3D representation called "Voint Cloud." The authors argue against existing heuristic-based aggregation, which could misleadingly incorporate representation from an arbitrary viewpoint. Then they attempted to address the research question of how to aggregate the per-view representation (in a multi-view setting) for a given 3D point cloud. In the process, they proposed their 3D representation, Voint, which is a view-dependent representation associated with each 3D point in a multi-view representation learning setting. The paper derived and described how their Voint Cloud (the proposed multi-view point cloud) could be used to learn representation end-to-end fashion to solve various 3D scene understanding tasks. Experimentally the authors demonstrated the usefulness of the proposed representation on three different 3D understanding tasks: i) 3D point cloud classification (on ScanObjectNN dataset), ii) 3D shape retrieval (on ScanNet Core55), and iii) 3D part segmentation (on ScanNet). The proposed method attained state-of-the-art performance in 3D point cloud classification and 3D object retrieval.

**Summary Of The Review:**

The paper has some weaknesses that have been discussed in the limitation section; however, it introduced a new idea (novel 3D representation) and a new methodology (VointNet and its corresponding operations).

---

> ### Author Response · Authors · 2022-11-18
> **rebuttal 1**
>
> ## P1. Outdoor 3D Scene Understanding
>
> In most of the literature, outdoor understanding (segmentation and 3D object detection) is usually addressed with different methods [c,d] than the ones used in indoor segmentation and detection [a,b]. This is due to the fact that outdoor datasets pose a different type of challenges than indoor ones (For example, point sparsity, and irregularity) that makes it impractical to apply the indoor methods to outdoors. Outdoor scans are usually sparse which affects the rendering quality and hence the quality of the 2D features.
>
> In our Voint cloud setup, the main rendering component requires dense point clouds in order to extract useful features. We studied the effect of the density of points on performance in Table 11 in the Appendix in Section C.2. which shows that even on dense point clouds, low density affects performance negatively.
>
> [a] Frustum pointnets for 3d object detection from rgb-d data , CVPR 2018
>
> [b] PointNeXt: Revisiting PointNet++ with Improved Training and Scaling Strategies, NeuRIPS 2022
>
> [c] Improving 3d object detection with channel-wise transformer, ICCV 2021
>
> [d] 2DPASS: 2D Priors Assisted Semantic Segmentation on LiDAR Point Clouds, ECCV 2022
>
> ## P2. Part Segmentation Results
>
> CurveNet indeed outperforms VointNet in the fixed setup but suffers in a rotated setup, which is more realistic. This is due to the fact that recent point-based methods are very good at local feature extraction based on coordinates and 3D neighborhoods, which gives an edge in fine-grained segmentation if the coordinates remain unchanged. Multi-view methods (including Voints) are better in global recognition, can account for changes in coordinates (including rotation), and incorporate more global information when making decisions, making them more robust than point-based methods. Yes, Table 4 VointNet is the MLP version as stated in Section 5 _Architectures_.
>
> ## P3. Newer 2D Backbones
> We agree with the reviewer that using newer and better 2D networks would likely improve performance in 3D as Table 6 suggests. This is a merit of the Voint cloud pipeline we propose. However, the only drawback of using large models in such a multi-view setup is that the computation cost tends to scale linearly with the number of views $M$. In Table 12 of Appendix Section C.2,  we report  the computation cost of each submodule in the pipeline. So, small 2D backbones tend to be favorable when tackling 3D tasks. This motivates more research in developing  efficient 2D models for use by the 3D community in the multi-view setup.
>
> ## P4. Number of views for Classification
> We thank the reviewer for the suggestion. Below are the results for classification accuracy on ModelNet40 on a different number of views ($M$). We add them in Table 13 in the updated Appendix Section C.2 as well.
>
> number of views (M): 4 - 6 - 8 - 10
>
> -----------------------------------------------------------------
>
> VointNet (Cls. Acc) : 90.3 , 90.8 , 92.0 , 92.3
>
> ## P5. Additional Classification Baselines
> Regarding KPConv, we report the classification results in Table 7 on ModelNet40 in the Appendix, in which it gets 92.9\% accuracy. This is comparable to the 92.8\% accuracy of VointNet. Other modern point-based classifiers like CurveNet [1] and PCT [2] are also now included in the same table in the revised Appendix. Additional baselines were added as well to Table 2 in the revised paper.
>
> ## P6. Adding Figures from Appendix
> We thank the reviewer for the suggestion. Due to space limitations, we moved these figures to Appendix.

---

### Author Response · Authors · 2022-11-18
**Thanks for the feedback**

We thank the reviewers for their insightful comments; we appreciate that they recognize our formulation's novelty, extensive experiments, and our paper presentation's quality.

---

### Decision · Program_Chairs · 2023-01-20

**Decision:**

Accept: poster

**Justification For Why Not Higher Score:**

While the work presents strong empirical results, it scope is limited to simpler synthetic object-centric settings -- a wider set of applications would have made this work stronger. Moreover, unlike recent trends in 3D processing where approaches aim to guarantee equi-variances to certain classes of transformations, this work may not be easy to adapt for such purposes.

**Justification For Why Not Lower Score:**

The paper presents an approach that is slightly unintuitive (and arguably not too elegant) but works surprisingly well, and is demonstrated across tasks. The community would benefit from seeing this approach for processing point clouds.

**Metareview: Summary, Strengths And Weaknesses:**

This work proposes to use multi-view representations for 3D point clouds and recommends first computing 2D features on renderings of a given point cloud, and then (via a learned module), aggregating these to obtain per-point representations. Empirical results across datasets and task show the generality and efficacy of the approach, and in particular demonstrates the benefits over 3D point cloud processing as well as prior multi-view methods with adhoc aggregation. The reviewers unanimously recommend acceptance, and the AC agrees.


**Note From Pc:**

if the above contains the word "oral" or "spotlight" please see: "oral" presentation means -> notable-top-5% and "spotlight" means -> notable-top-25%. As stated in our emails, we are disassociating presentation type from AC recommendations